# Learning List-Level Domain-Invariant Representations for Ranking

**Ruicheng Xian**[1†]    **Honglei Zhuang**[2]    **Zhen Qin**[2]    **Hamed Zamani**[3†]    **Jing Lu**[2]
**Ji Ma**[2]    **Kai Hui**[2]    **Han Zhao**[1]    **Xuanhui Wang**[2]    **Michael Bendersky**[2]

[1]University of Illinois Urbana-Champaign
`{rxian2,hanzhao}@illinois.edu`

[2]Google Research
`{hlz,zhenqin,ljwinnie,maji,kaihuibj,xuanhui,bemike}@google.com`

[3]University of Massachusetts Amherst
`zamani@cs.umass.edu`

## Abstract

Domain adaptation aims to transfer the knowledge learned on (data-rich) source domains to (low-resource) target domains, and a popular method is invariant representation learning, which matches and aligns the data distributions on the feature space. Although this method is studied extensively and applied on classification and regression problems, its adoption on ranking problems is sporadic, and the few existing implementations lack theoretical justifications. This paper revisits invariant representation learning for ranking. Upon reviewing prior work, we found that they implement what we call *item-level alignment*, which aligns the distributions of the items being ranked from all lists in aggregate but ignores their list structure. However, the list structure should be leveraged, because it is intrinsic to ranking problems where the data and the metrics are defined and computed on lists, not the items by themselves. To close this discrepancy, we propose *list-level alignment*—learning domain-invariant representations at the higher level of lists. The benefits are twofold: it leads to the first domain adaptation generalization bound for ranking, in turn providing theoretical support for the proposed method, and it achieves better empirical transfer performance for unsupervised domain adaptation on ranking tasks, including passage reranking.

## 1 Introduction

*Learning to rank* applies machine learning techniques to solve ranking problems that are at the core of many everyday products and applications, including search engines and recommendation systems [34]. The availability of ever-increasing amounts of training data and advances in modeling are constantly refreshing the state-of-the-art of many ranking tasks [40]. Yet, the training of practical and effective ranking models on tasks with little to no annotated data remains a challenge [57]. A popular transfer learning framework for this problem is *domain adaptation* [41]: given source domains with labeled data that are relevant to the target domain of interest, domain adaptation methods optimize the model on the source domain and make the knowledge transferable by utilizing (unlabeled) target data.

For domain adaptation, there are task-specific and general-purpose (unsupervised) methods. E.g., for text ranking in information retrieval where we only have access to target documents without any training query or relevance annotation, a method is to use generative language models to synthesize

---

[†]Work performed while at Google Research.

37th Conference on Neural Information Processing Systems (NeurIPS 2023).

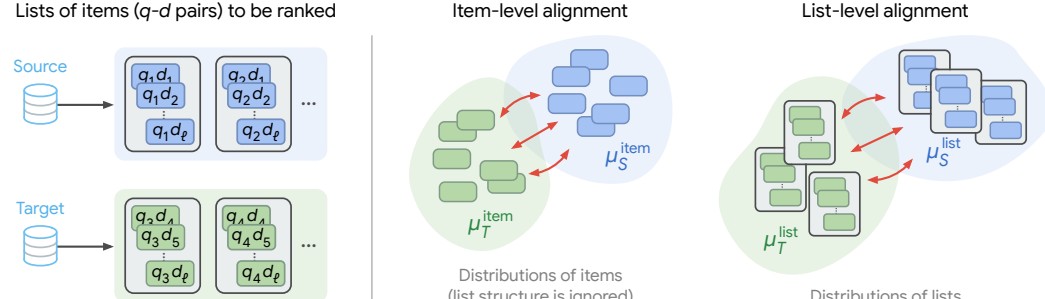

Figure 1: Item-level and list-level alignment on a ranking problem. Each list contains $\ell$ items to be ranked. Whereas item-level alignment aligns the distributions of items but ignores the list structure, list-level alignment preserves this structure and aligns the distributions of lists.

relevant queries for the unannotated documents, then train the ranking model on the synthesized query-document pairs [37, 54, 65]. For general-purpose methods, which are applicable to all kinds of tasks, perhaps the most well-known is *domain-invariant representation learning* [38, 17]. While it is studied extensively on classification and regression problems [71, 73], applications of invariant representation learning on ranking problems are sporadic [12, 58, 68], and the existing implementations lack theoretical justifications, which casts doubt on their applicability to learning to rank.

In this paper, we revisit the method of learning invariant representations for domain adaptation on ranking problems. We found that, from our review of prior work [12, 58, 68], existing implementations of the method all perform what we call *item-level alignment* (Fig. 1 and Section 3.2.1), which aligns the distributions of the items being ranked (e.g., query-document pairs) aggregated from all lists and discards their *list structure* (as in, e.g., q-d pairs with the same query all belong to the same list). However, both the data and the metrics for ranking are defined and computed on lists, not the items by themselves, so intuitively the list structure should be leveraged in the feature learning process as it is intrinsic to the problem. To close this discrepancy, we propose and analyze *list-level alignment*, which aligns the distributions of the lists and preserves their list structure (Section 3.2.2).

The conceptual leap from item-level to list-level alignment is significant. First, by analyzing list-level representation invariance, we establish the first domain adaptation generalization bound for ranking (Section 4), in turn providing theoretical support for our proposed method as well as a foundation for future studies. Second, list-level alignment provides empirical benefits to domain adaptation on ranking problems, which we demonstrate on two ranking tasks, passage reranking (Section 5) and Yahoo! LETOR web search ranking (Appendix D), where it outperforms zero-shot learning, item-level alignment, as well as a method specific to text ranking based on query synthesis [37].

## 2  Problem Setup

We define a ranking problem by a joint distribution $\mu$ of length-$\ell$ lists and their nonnegative scores, denoted by $X = (X_1, \cdots, X_\ell) \in \mathcal{X}$ and $Y = (Y_1, \cdots, Y_\ell) \in \mathbb{R}_{\geq 0}^\ell$.[1] For example, in information retrieval, the items in each list are query-document pairs of the same query, and the scores indicate the relevance of the documents to the query. Note that the lists are *permutation-invariant*, because switching the order of the items (and the corresponding scores) does not change the content of the list; concretely, let $S_\ell$ denotes the set of permutations on $[\ell] := \{1, 2, \cdots, \ell\}$, then permutation invariance means that $\mu((x_1, \cdots, x_\ell), (y_1, \cdots, y_\ell)) = \mu((x_{\pi_1}, \cdots, x_{\pi_\ell}), (y_{\pi_1}, \cdots, y_{\pi_\ell}))$ for all $(x, y)$ and $\pi \in S_\ell$. We assume that the scores are a function of the list, $Y = y(X)$, so the problem can be equivalently given by the marginal distribution $\mu^X$ of lists and a scoring function $y : \mathcal{X} \to \mathbb{R}_{\geq 0}^\ell$.

Learning to rank aims to learn a *ranker* that takes lists $x$ as input and outputs rank assignments $r \in S_\ell$, where $r_i \in [\ell]$ is the predicted rank of item $i$, such that $r$ recovers the descending order of the scores $y$, i.e., $y_i > y_j \iff r_i < r_j$ for all $i, j$. The more common setup, which we will adopt, is

---

[1]Although here the input lists are written to be decomposable into the $\ell$ items they contain, generally and more abstractly, they need not be so.

to train a *scorer*, $f : \mathcal{X} \to \mathbb{R}^\ell$, with the rank assignments obtained by first computing the ranking scores $s = f(x) \in \mathbb{R}^\ell$, where $s_i$ is the predicted score of item $i$, then taking the descending order of $s$ (alternatively, the rank assignments can be generated probabilistically; see Section 4).

To quantitatively measure the quality of the predicted ranks, we use (listwise) ranking metrics, $u : S_\ell \times \mathbb{R}^\ell_{\geq 0} \to \mathbb{R}_{\geq 0}$, which computes a utility score via comparing the predicted rank assignments to the ground-truth scores of the list. Common metrics in information retrieval include reciprocal rank and normalized discounted cumulative gain [63, 26]:

**Definition 2.1** (Reciprocal Rank). If the ground-truth scores are binary, i.e., $y \in \{0, 1\}^\ell$, then the reciprocal rank (RR) of a predicted rank assignments $r \in S_\ell$ is

$$\mathrm{RR}(r, y) = \max(\{r_i^{-1} : 1 \leq i \leq \ell, y_i = 1\} \cup \{0\}).$$

The average RR of a ranking model $f' : \mathcal{X} \to S_\ell$ on a dataset, $\mathbb{E}_{(X,Y) \sim \mu}[\mathrm{RR}(f'(X), Y)]$, is referred to as the mean reciprocal rank (MRR).

**Definition 2.2** (NDCG). The discounted cumulative gain (DCG) and the normalized DCG (with identity gain function) of a predicted rank assignments $r \in S_\ell$ are

$$\mathrm{DCG}(r, y) = \sum_{i=1}^{\ell} \frac{y_i}{\log(r_i + 1)}, \quad \text{and} \quad \mathrm{NDCG}(r, y) = \frac{\mathrm{DCG}(r, y)}{\mathrm{IDCG}(y)},$$

where $\mathrm{IDCG}(y) = \max_{r' \in S_\ell} \mathrm{DCG}(r', y)$, called the ideal DCG, is the maximum DCG of a list, which is attained by the descending order of $y$.

**Domain Adaptation.** In this learning setting, we have a source domain $(\mu_S^X, y_S)$ and a target domain $(\mu_T^X, y_T)$ with different data distributions,[2] and the goal is to learn a good ranking model for the target domain by leveraging all available resources: whereas access to source domain labeled training data is always assumed, we may only be given unlabeled data for the target domain (this scenario is called *unsupervised* domain adaptation). A popular method for this adaptation is domain-invariant representation learning, which matches and aligns the source and target domain data distributions on the feature space so that their distributions appear similar to the model.

## 3 Learning Domain-Invariant Representations for Ranking

We begin by reviewing invariant representation learning for domain adaptation and the optimization technique of adversarial training, then describe and compare two instantiations of this framework to ranking problems in Section 3.2—item-level alignment, which is implemented in prior work, and our proposed list-level alignment.

For representation learning, we train composite models of the form $f = h \circ g$, where $g : \mathcal{X} \to \mathcal{Z}$ is a shared feature map and $h$ is a task head on the shared features. As an example, if the model $f$ is end-to-end implemented as an $m$-layer multilayer perceptron (MLP), then we could treat the first $(m - 1)$ layers as $g$ and the last as $h$.

And, recall the definitions of Lipschitz continuity and Wasserstein distance [15]:

**Definition 3.1** (Lipschitz Continuity). A function $f : \mathcal{X} \to \mathcal{Y}$ from metric space $(\mathcal{X}, d_\mathcal{X})$ to $(\mathcal{Y}, d_\mathcal{Y})$ is $L$-Lipschitz, denoted by $f \in \mathrm{Lip}(L)$, if $d_\mathcal{Y}(f(x), f(x')) \leq L\, d_\mathcal{X}(x, x')$ for all $x, x' \in \mathcal{X}$.

**Definition 3.2** (Wasserstein Distance). The Wasserstein-1 distance between probability measures $p, q$ on metric space $\mathcal{X}$ is denoted and given by $W_1(p, q) = \sup_{\psi : \mathcal{X} \to \mathbb{R}, \psi \in \mathrm{Lip}(1)} \int_\mathcal{X} \psi(x)(p(x) - q(x))\, \mathrm{d}x$.

### 3.1 Invariant Representation Learning via Adversarial Training

Given a source domain $\mu_S$ and a target domain $\mu_T$, invariant representation learning aims to learn a shared feature map $g$ s.t. the distributions are aligned on the feature space, i.e., $D(\mu_S^Z, \mu_T^Z) \approx 0$, where $D$ is a divergence measure or probability metric, e.g., Wasserstein distance, and the source feature distribution $\mu_S^Z$ (analogously for target $\mu_T^Z$) is defined to be the distribution of $Z = g(X)$, $X \sim \mu_S^X$. The idea is that a composite model with an aligned feature representation is transferable between

---

[2] For adapting from multiple source domains, see [71].

domains—this is supported by the following generalization bound on classification problems [52, 5]. We will establish a similar result for ranking in Section 4.

**Theorem 3.3.** *Let binary classification problems on a source and a target domain be given by joint distributions $\mu_S, \mu_T$ of inputs and labels, $(X, Y) \in \mathcal{X} \times \{0, 1\}$. Define the error rate of a predictor $f : \mathcal{X} \to [0, 1]$ on $\mu$ by $\mathcal{R}(f) = \mathbb{E}_{(X,Y)\sim\mu}[f(X)\,\mathbb{1}[Y \neq 1] + (1 - f(X))\,\mathbb{1}[Y \neq 0]]$,[3] where $\mathbb{1}[\cdot]$ is the indicator function.*

*Let $\mathcal{H} \subset [0, 1]^{\mathcal{Z}}$ be an L-Lipschitz class of prediction heads, and for any $g \in \mathcal{G}$, define the minimum joint error rate by $\lambda_g^* = \min_{h'}(\mathcal{R}_S(h' \circ g) + \mathcal{R}_T(h' \circ g))$, then for all $h \in \mathcal{H}$,*

$$\mathcal{R}_T(h \circ g) \leq \mathcal{R}_S(h \circ g) + 2L\,W_1(\mu_S^Z, \mu_T^Z) + \lambda_g^*.$$

This result bounds the target domain risk of the composite model $h \circ g$ by its source task risk, plus the divergence of the feature distributions and the optimal joint risk. It therefore suggests that a good model for the target could be obtained by learning an informative domain-invariant feature map $g$ (s.t. $\lambda_g^*$ and $W_1(\mu_S^Z, \mu_T^Z) \approx 0$) and training $h$ to minimize the source task risk $\mathcal{R}_S$.

Invariant representation learning algorithms capture the aforementioned objectives with the joint training objective of $\min_{h,g}(\mathcal{L}_S(h \circ g) + \lambda\,D(\mu_S^Z, \mu_T^Z))$ [35, 17, 13], where $\mathcal{L}_S$ is the source task loss, and $\lambda > 0$ is a hyperparameter that controls the strength of invariant representation learning. This objective learns domain-invariant features by optimizing $g$ to minimize the distributional discrepancy, and minimizing source risk at the same time s.t. the learned features are useful for the task. Since it does not require labeled data but only unlabeled ones for estimating $W_1(\mu_S^Z, \mu_T^Z)$, it is applicable to unsupervised domain adaptation. If labeled target data are available, they can be incorporated by, e.g., adding a target loss $\mathcal{L}_T$, which will help keep $\lambda_g^*$ low.

**Adversarial Training.** When the divergence measure $D$ is simple and admits closed-form expressions, e.g., maximum-mean discrepancy [19], the objective above can be minimized directly [35], but they are typically too weak at modeling complex distributions that arise in real-world problems. To achieve feature alignment under stronger measures, e.g., JS divergence or Wasserstein distance, a well-known approach is adversarial training [18, 17, 2], which reformulates the above objective into

$$\mathcal{L}_{\text{joint}}(h, g) = \min_{h \in \mathcal{H}, g \in \mathcal{G}}\left(\mathcal{L}_S(h \circ g) - \lambda\min_{f_{\text{ad}} \in \mathcal{F}_{\text{ad}}}\mathcal{L}_{\text{ad}}(g, f_{\text{ad}})\right),$$

and optimizes it using gradient descent-ascent with a gradient reversal layer added on top of $g$ in the adversarial component.

The adversarial component, $-\min_{f_{\text{ad}}}\mathcal{L}_{\text{ad}}(g, f_{\text{ad}})$, can be shown to upper bound the divergence between $\mu_S^Z, \mu_T^Z$. It involves an adversary $f_{\text{ad}} : \mathcal{Z} \to \mathbb{R}$ (parameterized by neural networks) and an adversarial loss function $\ell_{\text{ad}} : \mathbb{R} \times \{0, 1\} \to \mathbb{R}$, whose inputs are the output of $f_{\text{ad}}$ and the domain identity $a$ (we set target domain to $a = 1$). The adversarial loss is computed over both domains:

$$\mathcal{L}_{\text{ad}}(g, f_{\text{ad}}) := \mathbb{E}_{X\sim\mu_S^X}[\ell_{\text{ad}}(f_{\text{ad}} \circ g(X), 0)] + \mathbb{E}_{X\sim\mu_T^X}[\ell_{\text{ad}}(f_{\text{ad}} \circ g(X), 1)].$$

If we choose the 0-1 loss, $\ell_{\text{ad}}(\hat{a}, a) = (1 - a)\,\mathbb{1}[\hat{a} \geq 0] + a\,\mathbb{1}[\hat{a} < 0]$, then the adversarial component upper bounds $W_1(\mu_S^Z, \mu_T^Z)$ (see Proposition A.1); here, $f_{\text{ad}}$ acts as a *domain discriminator* for distinguishing the domain identity, and $\mathcal{L}_{\text{ad}}$ is the balanced classification error of $f_{\text{ad}}$. For training, the 0-1 loss is replaced by a surrogate loss, e.g., the logistic loss [18], $\ell_{\text{ad}}(\hat{a}, a) = \log(1 + e^{(1-2a)\hat{a}})$.

## 3.2 Invariant Representation Learning for Ranking

This section describes and compares two instantiations of the invariant representation learning framework above for ranking: item-level alignment, and our proposed list-level alignment. The key difference between them is the choice of $\mu^Z$ whose divergence $W_1(\mu_S^Z, \mu_T^Z)$ is to be minimized.

**Model Setup.** We consider a composite scoring model $f = h \circ g$ where the feature map $g$ is s.t. given an input list $x = (x_1, \cdots, x_\ell)$, it outputs a list of $k$-dimensional feature vectors, $z = g(x) = (v_1, \cdots, v_\ell) \in \mathbb{R}^{\ell \times k}$, where $v_i \in \mathbb{R}^k$ corresponds to item $i$. Ranking scores are then

---

[3]This definition assumes that the output class is probabilistic according to $\mathbb{P}(\widehat{Y} = 1 \mid X = x) = f(x)$.

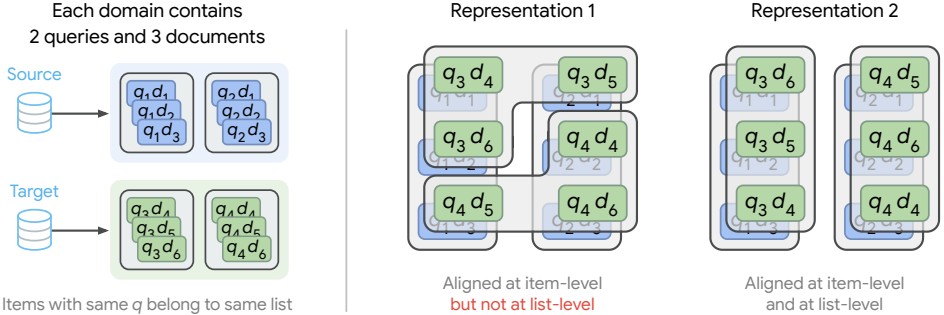

Figure 2: List-level alignment is a stronger requirement than item-level alignment, which, in addition to pairing target domain items to source items, it also pairs target lists to source lists.

obtained by, e.g., projecting each $v_i$ to $\mathbb{R}$ with a (shared) linear layer. This setup, which we will adopt in our experiments, is common with listwise ranking models and in many ranking systems [10]: in neural text ranking, each feature vector can be the embedding of the input text computed by a language model [39, 21, 76].

### 3.2.1 Item-Level Alignment

In item-level alignment (ItemDA), the distributions of the $k$-dimensional feature vectors from all lists in aggregate are aligned [12, 58, 68], i.e., $\mu_S^{Z,\text{item}} \approx \mu_T^{Z,\text{item}}$, where

$$\mu^{Z,\text{item}}(v) := \mathbb{P}_{Z \sim \mu^Z}(v \in Z) = \mathbb{P}_{X \sim \mu^X}(v \in g(X)), \quad \text{supp}(\mu^{Z,\text{item}}) \subseteq \mathbb{R}^k.$$

In other words (when both domain have the same number of lists), it finds a matching between the items of the source and target domains that pairs each target item with a source one (Fig. 2). Note that the list structure is not retained in this definition, because one cannot necessarily tell whether two items drawn from the bag of feature vectors, $v, v' \sim \mu^{Z,\text{item}}$, belong to the same list.

To implement item-level alignment, the discriminator $f_{\text{ad}}$ (usually an MLP) is set up to take individual feature vectors $v \in \mathbb{R}^k$ as input. In each forward-pass, the feature map $g$ computes a batch of feature vector lists, $\{z_i\}_{i \in [b]} = \{(v_{i1}, \cdots, v_{i\ell})\}_{i \in [b]}$, then the discriminator takes the items batched from all lists (this step discards the list structure), $\{v_{ij}\}_{i \in [b], j \in [\ell]}$, and predicts their domain identities.

Item-level alignment is identical to invariant representation learning implementations for classification and regression, e.g., DANN [17], which also operate on "items", since neither the data nor the metrics in those problem settings have any explicit structural assumptions. However, ranking problems are inherently defined with list structures: the inputs to the model are organized by lists, and ranking metrics are evaluated on lists rather than the items by themselves. This discrepancy casts doubt on the applicability of item-level alignment to ranking problems, and moreover, it is unclear whether domain adaptation for ranking via item-level alignment is justified from a theoretical perspective.

### 3.2.2 List-Level Alignment

Closing the discrepancy discussed above, we propose list-level alignment (ListDA), which aligns the distributions of the lists of feature vectors on the $\mathbb{R}^{\ell \times k}$ space, i.e., $\mu_S^{Z,\text{list}} \approx \mu_T^{Z,\text{list}}$, where

$$\mu^{Z,\text{list}}(z) := \mathbb{P}_{Z \sim \mu^Z}(z) = \mathbb{P}_{X \sim \mu^X}(z = g(X)), \quad \text{supp}(\mu^{Z,\text{list}}) \subseteq \mathbb{R}^{\ell \times k}.$$

This means finding a matching that not only pairs all source and target items, but also pairs items from the same target list to items from the same source list. Therefore, list-level alignment is a stronger requirement than item-level alignment, $\mu_S^{Z,\text{list}} = \mu_T^{Z,\text{list}} \implies \mu_S^{Z,\text{item}} = \mu_T^{Z,\text{item}}$, while the converse is generally not true (see Fig. 2 for a picture, or Example A.2)!

To implement list-level alignment, the discriminator has to make predictions on lists, $z \in \mathbb{R}^{\ell \times k}$, i.e., in each forward-pass it would be fed a batch of feature vector lists, $\{z_i\}_{i \in [b]} = \{(v_{i1}, \cdots, v_{i\ell})\}_{i \in [b]}$, and output $b$ predictions. For this, a possible design choice is to flatten each feature vector list into a long $(\ell k)$-dimensional vector and use an MLP as $f_{\text{ad}}$. But recall from Section 2 that the

input lists are permutation-invariant, and so are the feature vector lists. Since the above setup does not incorporate this property and an exponential amount of samples is required to see all possible permutations, this would be inefficient to optimize; yet, the optimality of $f_{\text{ad}}$ is essential to the success of adversarial training. So instead, we use transformers with mean-pooling less positional encoding as the discriminator [60], which is permutation-invariant to the input.

Compared to item-level alignment, our list-level alignment is supported by a domain adaptation generalization bound (Theorem 4.7, established in the next section), and achieves better transfer performance in our Section 5 and Appendix D evaluations. These results suggest that the list structure is essential for an effective domain adaptation on ranking problems.

## 4 Generalization Bound for Ranking with List-Level Alignment

Based on the list-level alignment proposed above, we establish a domain adaptation generalization bound for ranking by analyzing list-level representation invariance. The discussions in this section consider learning a composite scoring model, $f = h \circ g : \mathcal{X} \to \mathbb{R}^\ell$, but need not assume that the list feature representation $z = g(x)$ can be decomposed into item-level feature representations as in the model setup of Section 3.2.2, only that it resides in a metric space $\mathcal{Z}$.

Our result can be viewed as an instantiation of the framework established originally for classification by Ben-David et al. [5]. But since ranking is a very different task with its unique technical challenges, our analysis differs from those in prior work [52]. We will introduce appropriate assumptions for handling these difficulties leading up to the main Theorem 4.7, which the readers may skip to.

The first hurdle is the discontinuity of the sorting operation. So rather than taking the descending order of the predicted scores to obtain the rank assignments, we generate them probabilistically using a *Plackett-Luce model* [43, 36] with the exponentiated scores as its parameters [10, 20]. This makes the computation of the utility (Lipschitz) continuous w.r.t. the raw output scores of the model.

**Definition 4.1** (Plackett-Luce Model). A Plackett-Luce (P-L) model with parameters $w \in \mathbb{R}^\ell_{>0}$ specifies a distribution over $S_\ell$, whose probability mass function $p_w$ is

$$p_w(r) = \prod_{i=1}^{\ell} \frac{w_{I(r)_i}}{\sum_{j=i}^{\ell} w_{I(r)_j}}, \quad \forall r \in S_\ell,$$

where $I(r)_i$ is the index of the item with rank $i$, $r_{I(r)_i} = i$.

**Assumption 4.2.** Given predicted scores $s = f(x) \in \mathbb{R}^\ell$, we generate the rank assignments probabilistic from a P-L model by $R \sim p_{\exp(s)} \in S_\ell$, where $\exp$ is taken coordinate-wise.

Under this assumption, the utility of a scorer $f$ on $(\mu^X, y)$ w.r.t. ranking metric $u : S_\ell \times \mathbb{R}^\ell_{\geq 0} \to \mathbb{R}_{\geq 0}$ is computed by (overloading $u$ for future brevity)

$$\mathbb{E}_\mu[u(f)] := \mathbb{E}_{X \sim \mu^X} \mathbb{E}_{R \sim p_{\exp(f(X))}}[u(R, y(X))].$$

We define the risk (or suboptimality) of $f$ as the difference between its utility to the maximum-attainable utility on the problem (e.g., the maximum is 1 when $u = $ NDCG):

$$\mathcal{R}(f) = \mathbb{E}_{X \sim \mu^X}\left[\max_{r \in S_\ell} u(r, y(X))\right] - \mathbb{E}_\mu[u(f)].$$

The next technical challenge is that unlike in classification where the perfect classifier is unique (i.e., achieving zero error), the perfect ranker is generally not: e.g., both rank assignments $(1, 2, 3)$ and $(2, 1, 3)$ achieve maximum utility (i.e., $\mathcal{R} = 0$) on a list with ground-truth scores $(1, 1, 0)$. Prior analyses of domain adaptation leverage this uniqueness [5, 52]; instead, ours does not rely on uniqueness (else the bound would be loose) with the following Lipschitz assumptions:

**Assumption 4.3.** The ranking metric $u : S_\ell \times \mathbb{R}^\ell_{\geq 0} \to \mathbb{R}_{\geq 0}$ is upper bounded by $B$, and is $L_u$-Lipschitz w.r.t. the ground-truth scores $y$ in the second argument (in Euclidean distance).

**Assumption 4.4.** The input lists $X$ reside in a metric space $\mathcal{X}$, and the ground-truth scoring function $y : \mathcal{X} \to \mathbb{R}^\ell_{\geq 0}$ is $L_y$-Lipschitz (in Euclidean distance on the output space).

We will show that Assumption 4.3 is satisfied by both RR and NDCG. Assumption 4.4 says that similar lists (i.e., close in $\mathcal{X}$) should have similar ground-truth scores, and is satisfied, e.g., when $\mathcal{X}$ is finite; such is the case with text data, which are one-hot encoded after tokenization.

**Assumption 4.5.** The list features $Z$ reside in a metric space $\mathcal{Z}$, and the class $\mathcal{H}$ of scoring functions, $h : \mathcal{Z} \to \mathbb{R}^\ell$, is $L_h$-Lipschitz (in Euclidean distance on the output space).

**Assumption 4.6.** The class $\mathcal{G}$ of feature maps, $g : \mathcal{X} \to \mathcal{Z}$, satisfies that $\forall g \in \mathcal{G}$, the restrictions of $g$ to the supports of $\mu_S^X$ and $\mu_T^X$, $g|_{\mathrm{supp}(\mu_S^X)}, g|_{\mathrm{supp}(\mu_T^X)}$ respectively, are both invertible with $L_g$-Lipschitz inverses.

Assumption 4.5 is standard in generalization and complexity analyses, and could be enforced, e.g., with $L^2$-regularization [1, 4]. The last assumption is technical, saying that the inputs can be recovered from their feature representations by a Lipschitz inverse $g^{-1}$. This means that the feature map $g$ should retain as much information from the inputs (on each domain), which is a desideratum of representation learning. Note that this assumption does not conflict with the goal of invariant representation learning: there is always a sufficiently expressive $\mathcal{G}$ s.t. $\exists g \in \mathcal{G}$ satisfying the assumption and $\mu_S^{Z,\mathrm{list}} = \mu_T^{Z,\mathrm{list}}$.

We are now ready to state our domain adaptation generalization bound for ranking:

**Theorem 4.7.** *Under Assumptions 4.2 to 4.6, for any $g \in \mathcal{G}$, define the minimum joint risk by* $\lambda_g^* = \min_{h'}(\mathcal{R}_S(h' \circ g) + \mathcal{R}_T(h' \circ g))$, *then for all $h \in \mathcal{H}$,*

$$\mathcal{R}_T(h \circ g) \leq \mathcal{R}_S(h \circ g) + 4(L_u L_y L_g + B\ell L_h) \, W_1(\mu_S^{Z,\mathrm{list}}, \mu_T^{Z,\mathrm{list}}) + \lambda_g^*,$$

*where $\mu^{Z,\mathrm{list}}$ is the marginal distribution of the list features $Z = g(X)$, $\mu^{Z,\mathrm{list}}(z) := \mu^X(g^{-1}(z))$.*

The key feature of this bound is that it depends on list-level alignment, $W_1(\mu_S^{Z,\mathrm{list}}, \mu_T^{Z,\mathrm{list}})$, hence it provides theoretical support for the list-level alignment proposed in Section 3.2.2. It can be instantiated to specific ranking metrics by simply verifying the Lipschitz condition $L_u$ of Assumption 4.3:

**Corollary 4.8** (Bound for MRR). RR *is 1-Lipschitz in $y$, thereby*

$$\mathbb{E}_{\mu_T}[\mathrm{RR}(h \circ g)] \geq \mathbb{E}_{\mu_S}[\mathrm{RR}(h \circ g)] - 4(L_y L_g + \ell L_h) \, W_1(\mu_S^{Z,\mathrm{list}}, \mu_T^{Z,\mathrm{list}}) - \lambda_g^*.$$

**Corollary 4.9** (Bound for NDCG). *If $C^{-1} \leq \mathrm{IDCG} \leq C$ for some $C \in (0, \infty)$ on $(\mu_S^X, y_S)$ and $(\mu_T^X, y_T)$ almost surely,[4] then* NDCG *is $\widetilde{O}(C\sqrt{\ell})$-Lipschitz in $y$ almost surely, thereby*

$$\mathbb{E}_{\mu_T}[\mathrm{NDCG}(h \circ g)] \geq \mathbb{E}_{\mu_S}[\mathrm{NDCG}(h \circ g)] - \widetilde{O}(C\sqrt{\ell}L_y L_g + \ell L_h) \, W_1(\mu_S^{Z,\mathrm{list}}, \mu_T^{Z,\mathrm{list}}) - \lambda_g^*.$$

## 5 Experiments on Passage Reranking

To demonstrate the empirical benefits of invariant representation learning with list-level alignment (ListDA) for unsupervised domain adaptation, we evaluate it on the passage reranking task[5] and compare to zero-shot transfer, item-level alignment (ItemDA), as well as a recent method based on query synthesis [37]. Note that our method is not specialized to text (re)ranking; in Appendix D, we evaluate ListDA on a web ranking task from the Yahoo! Learning to Rank Challenge.

In passage reranking, given a text query, we are to rank candidate passages based on their relevance to the query from a retrieved set. Reranking is employed in scenarios where the corpus is too large for using accurate but expensive models such as cross-attention neural rankers to rank every (millions of) documents. Rather, a two-stage approach is taken: a simple but efficient model such as sparse or dense retrievers (e.g., BM25 [49] and DPR [28]) is first used to retrieve a candidate set of (hundreds of) passages, then a more sophisticated (re)ranking model is used to refine and improve their ranks.

We use BM25 to retrieve 1,000 candidate passages in the first-stage, and focus on the adaptation of a RankT5 listwise reranker [76], a cross-attention model derived from the T5 base language model with 250 million parameters [48]. It takes the concatenation of the query and the document (q-d pair) as input and outputs an embedding. We treat the q-d embeddings computed on the same list as the list feature—consistent with the setting in Section 3.2. For the task training loss function, we use the softmax cross-entropy ranking loss: $\ell(s, y) = -\sum_{i=1}^{\ell} y_i \log(\exp(s_i)/\sum_{j=1}^{\ell} \exp(s_j))$.

---

[4]IDCG is lower bounded, e.g., if every list contains at least one relevant item, and upper bounded when the ground-truth scores are upper bounded.

[5]We will use the terms *document* and *passage* interchangeably.

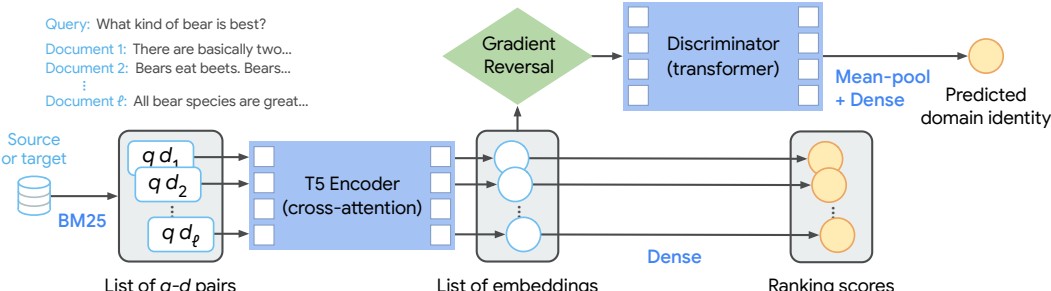

Figure 3: Block diagram of the cross-attention RankT5 text ranking model with ListDA.

**Datasets.** In our domain adaptation experiments, we use the MS MARCO dataset for passage ranking [3] as the source domain, which contains 8 million passages crawled from the web covering a wide range of topics with 532,761 search query and relevant passage pairs. The target domains are biomedical (TREC-COVID [62], BioASQ [59]) and news articles (Robust04 [64]). The data are preprocessed consistently with the BEIR benchmark [57]; their paper includes dataset statistics.

Since the training split of the target domain datasets does not contain q-d relevance annotations, our setting is unsupervised. It also does not contain any queries, but these are required for performing domain adaptation, because the input lists are defined to consist of q-d pairs. To acquire training queries on the target domains, we synthesize them in a zero-shot manner following [37], by using a T5 XL query generator (QGen) trained on MS MARCO relevant q-d pairs. QGen synthesizes each query as a seq-to-seq task given a passage, and the generated queries are *expected* to be relevant to the input passages. See Table 7 for sample QGen q-d pairs.

**Methods.**[6] In **zero-shot** transfer, the reranker is trained on MS MARCO and evaluated on the target domain without adaptation. In **QGen pseudolabel** (QGen PL), we treat QGen q-d pairs synthesized on the target domain as relevant pairs, and train the reranker on them in addition to MS MARCO. This method is specific to text ranking, and is used in several recent works on domain adaptation of text retrievers and rerankers [37, 54, 65].

**ItemDA** (item-level alignment) is the prior implementation of invariant representation learning for ranking [12, 58, 68], and is set up according to Section 3.2.1 with adversarial training described in Section 3.1. The adversarial loss is aggregated from the losses of five three-layer MLP discriminators (no improvements from using wider or more layers), $\sum_{i=1}^{5} \mathcal{L}_{\text{ad}}(g, f_{\text{ad}}^{(i)})$, for reducing the sensitivity to the randomness in the initialization and training process [16]. Our **ListDA** (list-level alignment) is set up according to Section 3.2.2, and the discriminator is an ensemble of five stacks of three T5 (transformer) encoding blocks. To predict the domain identity of a list feature $z = (v_1, \cdots, v_\ell)$, we feed the vectors into the transformer blocks all at once as a sequence, take the mean-pool of the $\ell$ output embeddings, then project it to a logit with a linear layer. A block diagram of ListDA is in Fig. 3.

Further details including hyperparameters and the construction of training lists (i.e., sampling negative pairs) are relegated to Appendix C, where we also include additional results such as using the pairwise logistic ranking loss, a hybrid adaptation method combining ListDA and QGen PL, and using transformers as the discriminator for ItemDA.

## 5.1 Results

The main results are presented in Table 1, where we report metrics that are standard in the literature (e.g., TREC-COVID uses NDCG@20), and evaluate rank assignments by the descending order of the predicted scores. Since TREC-COVID and Robust04 are annotated with 3-level relevancy, the scores are binarized for mean average precision (MAP) and MRR as follows: on TREC-COVID, we map 0 (not relevant) and 1 (partially relevant) to negative, and 2 (fully relevant) to positive; on Robust04, 0 (not relevant) to negative, and 1 (relevant) and 2 (highly relevant) to positive.

---

[6]All adaptation methods are applied on each source-target domain pair separately.

Table 1: Reranking performance of RankT5 on top 1000 BM25-retrieved passages.

| Target domain | Method | MAP | MRR@10 | NDCG@5 | NDCG@10 | NDCG@20 |
|---|---|---|---|---|---|---|
| Robust04 | BM25 | 0.2282 | 0.6801 | 0.4396 | 0.4088 | 0.3781 |
| | Zero-shot | 0.2759 | 0.7977$^\dagger$ | 0.5857$^\dagger$ | 0.5340$^\dagger$ | 0.4856$^\dagger$ |
| | QGen PL | 0.2693 | 0.7644 | 0.5406 | 0.5034 | 0.4694 |
| | ItemDA | 0.2822$^{*\dagger}$ | 0.8037$^\dagger$ | 0.5822$^\dagger$ | 0.5396$^\dagger$ | 0.4922$^\dagger$ |
| | ListDA | **0.2901**$^{*\dagger\ddagger}$ | **0.8234**$^{*\dagger}$ | **0.5979**$^{\dagger\ddagger}$ | **0.5573**$^{*\dagger\ddagger}$ | **0.5126**$^{*\dagger\ddagger}$ |
| TREC-COVID | BM25 | 0.2485 | 0.8396 | 0.7163 | 0.6559 | 0.6236 |
| | Zero-shot | 0.3083 | 0.9217 | 0.8328 | 0.8200 | 0.7826 |
| | QGen PL | 0.3180$^{*\ddagger}$ | 0.8907 | 0.8373 | 0.8118 | 0.7861 |
| | ItemDA | 0.3087 | 0.9080 | 0.8276 | 0.8142 | 0.7697 |
| | ListDA | **0.3187**$^{*\ddagger}$ | **0.9335** | **0.8693**$^{*\ddagger}$ | **0.8412**$^{\dagger\ddagger}$ | **0.7985**$^\ddagger$ |
| BioASQ | BM25 | 0.4088 | 0.5612 | 0.4580 | 0.4653 | 0.4857 |
| | Zero-shot | 0.5008$^\ddagger$ | 0.6465 | 0.5484$^\ddagger$ | 0.5542$^\ddagger$ | 0.5796$^\ddagger$ |
| | QGen PL | 0.5143$^{*\ddagger}$ | 0.6551 | 0.5538$^\ddagger$ | 0.5643$^\ddagger$ | 0.5915$^{*\ddagger}$ |
| | ItemDA | 0.4781 | 0.6383 | 0.5315 | 0.5343 | 0.5604 |
| | ListDA | **0.5191**$^{*\ddagger}$ | **0.6666**$^{*\ddagger}$ | **0.5639**$^{*\ddagger}$ | **0.5714**$^{*\ddagger}$ | **0.5985**$^{*\ddagger}$ |

Source domain is MS MARCO. Gain function in NDCG is the identity map. $^*$Improves upon zero-shot with statistical significance ($p \leq 0.05$) under the two-tailed Student's $t$-test. $^\dagger$Improves upon QGen PL. $^\ddagger$Improves upon ItemDA.

Across all three datasets, ListDA achieves the best performance, and the fact that it uses the same training resource as QGen PL demonstrates the added benefits of list-level invariant representations for domain adaptation. The favorable comparison of ListDA to ItemDA corroborates the discussion in Sections 3.2 and 4 that for domain adaptation on ranking problems, item-level alignment is insufficient for transferability, sometimes even resulting in negative transfer (vs. zero-shot). Rather, list-level alignment is the more appropriate approach.

## 5.2 Analyses of ListDA

**Quality of QGen.** An explanation for why ListDA outperforms QGen PL despite the same resources is that the sampling procedure (Appendix C.3) of negative q-d pairs could introduce false negatives into the training data. This is supported by the observation in [54] that QGen synthesized queries lack specificity and could be relevant to many documents. While these false negative labels are used for training in QGen PL, they are discarded in ListDA, which is thereby less likely to be affected by false negatives, or even false positives—when synthesized queries are not relevant to the input passages (see Table 9 for samples). Although out of the scope, better query generation is expected to improve the performance of both QGen PL and ListDA.

**Size of Target Data.** Unsupervised domain adaptation requires sufficient unlabeled data, but not all domains have the same amount: BioASQ has 14 million documents (so is the total number of QGen queries, as we synthesize one per document), but Robust04 only has 528,155, and TREC-COVID 171,332.

In Fig. 4, we plot the performance of ListDA under varying numbers of target QGen queries (which is the number of target training lists). Surprisingly, on Robust04 and TREC-COVID, using just ~100 target QGen queries (0.03% and 0.06% of all, respectively) is sufficient for ListDA to achieve full performance! Although the number of queries is small, since we retrieve 1,000 documents per query, the amount of distinct target documents in those 100 lists could be substantial—up to 100,000, or 29.5% and 60.7% of the respective corpora.

The performance begins to drop when reduced to ~10 queries, which caps the amount of distinct documents at 10,000 (2.7% and 5.8%, respectively). The same data efficiency, however, is

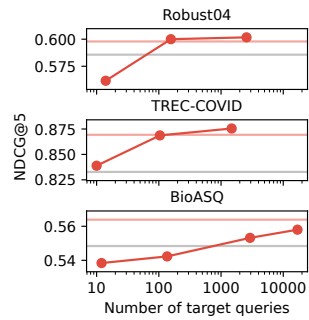

Figure 4: ListDA performance under different target sizes. Lower grey horizontal line is zero-shot, upper red line is ListDA using all QGen queries.

not observed on BioASQ, likely due to the size and hardness of the dataset, such as the use of specialized biomedical terms (see examples in Tables 7 to 9).

# 6   Related Work

**Learning to Rank and Text Ranking.**   Learning to rank historically focuses on tabular data, for which, a wide range of models are developed [34], from SVMs [27], gradient boosted decision trees [9], to neural rankers [8, 42, 45]. Another traditional research direction is the design of ranking losses (surrogate to ranking metrics), which can be categorized into pointwise, pairwise, and listwise approaches [10, 7, 74, 24].

Recent advances in large neural language models have spurred interest in applying them on text ranking tasks [33, 46], leading to cross-attention models [22, 39, 40, 44] and generative models based on query likelihood [14, 77, 78, 50]. A different line of work is neural text retrieval models, which emphasizes efficiency, and has seen the development of dual-encoder [28, 70], late-interaction [29, 23], and models leveraging transformer memory [56].

**Domain Adaptation in Information Retrieval.**   Work on this subject can be categorized into supervised and unsupervised domain adaptation. The former assumes access to labeled source data and (a small amount of) labeled target data (a.k.a. few-shot learning) [54]. This work focuses on the unsupervised setting, where target domain data do not have annotations. Cohen et al. [12] apply invariant representation learning to unsupervised domain adaptation for text ranking, followed by Tran et al. [58] for enterprise email search, and Xin et al. [68] for dense retrieval. However, unlike our proposed list-level alignment method (ListDA), they learn invariant representations at the item-level. A recent family of adaptation methods is based on query generation [37, 65], originally proposed for dense retrieval.

**Invariant Representation Learning.**   Domain-invariant representation learning is a popular family of adaptation methods [35, 17, 13], to which our proposed ListDA belongs. Besides ranking, these methods are also applied in fields including vision and language, and on tasks ranging from cross-domain sentiment analysis, question-answering [31, 61], and to cross-lingual learning and machine translation [67, 30].

Zhao et al. [72] and Tachet des Combes et al. [55] point out that on classification problems, attaining perfect feature alignment and high source accuracy are insufficient to guarantee good target performance. This occurs when the marginal distributions of labels differ, or the learned features contain domain-specific and nontransferable components. Although their findings do not directly apply to ranking, still, they suggest two potential directions for future work: one is whether Theorem 4.7 would admit a fundamental lower bound under distributional shifts in the ground-truth scores, and another is to modify ListDA to include a component for isolating out nontransferable features, as in domain separation networks [6].

# 7   Conclusion

We proposed an implementation of invariant representation learning for ranking via list-level feature alignment, and based on which, established a domain adaptation generalization bound for ranking. Our theoretical and empirical results illustrate the significance of preserving the list structure for achieving effective domain adaptation, which is overlooked in prior work.

A broader message is that when working with (feature) representations, either for domain adaptation or other purposes, they should be analyzed at the same level (or structure) at which the data are defined for the task of interest and the metrics are computed. The results of this paper is such an example—by moving from item-level alignment to the more appropriate list-level alignment, we extracted more potentials from invariant representation learning for domain adaptation on ranking problems.

## Acknowledgments

We thank the anonymous reviewers for their comments and suggestions on improving the presentation. This research was supported in part by the Google Visiting Researcher program and in part by the Center for Intelligent Information Retrieval. Han Zhao's work was partly supported by the Defense Advanced Research Projects Agency (DARPA) under Cooperative Agreement Number: HR00112320012. Any opinions, findings, and conclusions or recommendations expressed in this material are those of the authors and do not necessarily reflect those of the sponsors.

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

# A Proofs for Section 3

**Proposition A.1.** *Let $\mu, \mu'$ be distributions supported on a metric space $(\mathcal{X}, d)$, define and assume $R = \sup_{(x,x') \in \mathrm{supp}(\mu \times \mu')} d(x, x') \leq \infty$, and let $\mathcal{L}(f)$ denote the balanced 0-1 loss of a discriminator $f : \mathcal{X} \to \{0, 1\}$, given by*

$$\mathcal{L}(f) := \mathbb{E}_{X \sim \mu}[f(X)] + \mathbb{E}_{X' \sim \mu'}[1 - f(X')],$$

*then $W_1(\mu, \mu') \leq R(1 - \min_f \mathcal{L}(f))$.*

*Proof.* Let $\Gamma(\mu, \mu')$ denote the collection of couplings between $\mu, \mu'$, then by the definition of the Wasserstein-1 distance,

$$\begin{aligned}
W_1(\mu, \mu') &= \inf_{\gamma \in \Gamma(\mu, \mu')} \int_{\mathcal{X} \times \mathcal{X}} d(x, x') \, \mathrm{d}\gamma(x, x') \\
&\leq R \inf_{\gamma \in \Gamma(\mu, \mu')} \int_{\mathcal{X} \times \mathcal{X}} \mathbb{1}[x \neq x'] \, \mathrm{d}\gamma(x, x') \\
&= R \left( 1 - \sup_{\gamma \in \Gamma(\mu, \mu')} \int_{\mathcal{X} \times \mathcal{X}} \mathbb{1}[x = x'] \, \mathrm{d}\gamma(x, x') \right) \\
&= R \left( 1 - \int_{\mathcal{X}} \min(\mu(x), \mu'(x)) \, \mathrm{d}x \right) \\
&= R \int_{\mathcal{X}} \max(0, \mu'(x) - \mu(x)) \, \mathrm{d}x \\
&= \frac{R}{2} \int_{\mathcal{X}} |\mu'(x) - \mu(x)| \, \mathrm{d}x,
\end{aligned} \tag{1}$$

because $\int (\mu'(x) - \mu(x)) \, \mathrm{d}x = 0$. Note that the last line is $R$ times the total variation between $\mu, \mu'$. On the other hand, rewrite

$$\mathcal{L}(f) = \int_{\mathcal{X}} (f(x)\mu(x) + (1 - f(x))\mu'(x)) \, \mathrm{d}x = 1 - \int_{\mathcal{X}} \left( \frac{1}{2} - f(x) \right) (\mu(x) - \mu'(x)) \, \mathrm{d}x.$$

We know that the minimum 0-1 loss is attained by the Bayes predictor, $f^*(x) = \mathbb{1}[\mu'(x) \geq \mu(x)]$, so

$$\min_f \mathcal{L}(f) = \mathcal{L}(f^*) = 1 - \frac{1}{2} \int_{\mathcal{X}} |\mu(x) - \mu'(x)| \, \mathrm{d}x \leq 1 - \frac{1}{R} W_1(\mu, \mu') \tag{2}$$

by Eq. (1). The result then follows from an algebraic rearrangement of terms in Eq. (2). $\qquad \square$

Next, we provide a concrete example (supplementing the picture in Fig. 2) for showing that item-level alignment does not necessarily imply list-level alignment.

*Example* A.2. Consider two uniform distributions $\mu_S^{\text{list}}, \mu_T^{\text{list}}$ of two lists containing three real-valued items, where the lists are

$$\mathrm{supp}(\mu_S^{\text{list}}) = \{(1, 2, 3), (4, 5, 6)\}, \quad \mathrm{supp}(\mu_T^{\text{list}}) = \{(1, 3, 5), (2, 4, 6)\}.$$

The item-level distributions, $\mu_S^{\text{item}}, \mu_T^{\text{item}}$, which are obtained by aggregating the items from all lists, are both the uniform distribution of the same six items:

$$\mathrm{supp}(\mu_S^{\text{item}}) = \mathrm{supp}(\mu_T^{\text{item}}) = \{1, 2, 3, 4, 5, 6\}.$$

Note that item-level representations are automatically aligned since $\mu_S^{\text{item}} = \mu_T^{\text{item}}$, but the list-level representations are not, because $\mathrm{supp}(\mu_S^{\text{list}}) \cap \mathrm{supp}(\mu_T^{\text{list}}) = \emptyset$.

# B Proofs for Section 4

This section provides the proof to Theorem 4.7, the domain adaptation generalization bound for ranking with list-level alignment. First, recall the following properties of Lipschitz functions:

**Fact B.1** (Properties of Lipschitz Functions).

1. *If $f : \mathbb{R}^d \to \mathbb{R}$ is differentiable, then it is L-Lipschitz (in Euclidean distance) if and only if $\|\nabla f\|_2 \le L$.*

2. *If $f : \mathcal{X} \to \mathbb{R}$ is L-Lipschitz and $g : \mathcal{X} \to \mathbb{R}$ is M-Lipschitz, then $(af + bg) \in \mathrm{Lip}(|a|L + |b|M)$, and $\max(f, g) \in \mathrm{Lip}(\max(L, M))$.*

3. *If $f : \mathcal{X} \to \mathcal{Y}$ is L-Lipschitz and $g : \mathcal{Y} \to \mathcal{Z}$ is M-Lipschitz, then $g \circ f \in \mathrm{Lip}(LM)$.*

*Proof.*

1. For the backward direction, suppose bounded gradient norm, $\|\nabla f\|_2 \le L$, then by the mean value theorem, $\exists t \in [0, 1]$ s.t. $f(y) - f(x) = \nabla f(z)^\top (y - x)$ with $z := (1 - t)x + ty$, so by Cauchy-Schwarz inequality,
$$\|f(y) - f(x)\|_2 \le \|\nabla f(z)\|_2 \|y - x\|_2 \le L\|y - x\|_2.$$

   For the forward direction, suppose $f \in \mathrm{Lip}(L)$, then by differentiability, $\nabla f(x)^\top z = f(x + z) - f(x) + o(\|z\|_2)$. Set $z := t\nabla f(x)$, we have
$$t\|\nabla f(x)\|_2^2 = f(x + t\nabla f(x)) - f(x) + o(t\nabla f(x)) \le Lt\|\nabla f(x)\|_2 + o(t\|\nabla f(x)\|_2),$$
   and the result follows by dividing both sides by $t\|\nabla f(x)\|_2$ and taking $t \to 0$.

2. First,
$$|af(x) + bg(x) - (af(y) + bg(y))| \le |a||f(x) - f(y)| + |b||g(x) - g(y)|$$
$$\le (|a|L + |b|M)\, d_{\mathcal{X}}(x, y).$$
   Next, assume w.l.o.g. $\max(f(x), g(x)) - \max(f(y), g(y)) \ge 0$, then
$$|\max(f(x), g(x)) - \max(f(y), g(y))|$$
$$= \begin{cases} f(x) - \max(f(y), g(y)) \le f(x) - f(y) \le L\, d_{\mathcal{X}}(x, y) & \text{if } f(x) \ge g(x) \\ g(x) - \max(f(y), g(y)) \le g(x) - g(y) \le M\, d_{\mathcal{X}}(x, y) & \text{else} \end{cases}$$
$$\le \max(L, M)\, d_{\mathcal{X}}(x, y).$$

3. $d_{\mathcal{Z}}(g \circ f(x), g \circ f(y)) \le M\, d_{\mathcal{Y}}(f(x), f(y)) \le LM\, d_{\mathcal{X}}(x, y).$ $\qquad\square$

We first prove Theorem 3.3 as a warm-up (restated below), the domain adaptation generalization bound for binary classification [52], since its proof shares the same organization as that of our Theorem 4.7, and it helps familiarizing readers of prior domain adaptation results and analysis techniques.

**Theorem B.2.** *Let binary classification problems on a source and a target domain be given by joint distributions $\mu_S, \mu_T$ of inputs and labels, $(X, Y) \in \mathcal{X} \times \{0, 1\}$. Define the error rate of a predictor $f : \mathcal{X} \to [0, 1]$ on $\mu$ by*
$$\mathcal{R}(f) = \mathbb{E}_{(X,Y)\sim\mu}[f(X)\,\mathbb{1}[Y \ne 1] + (1 - f(X))\,\mathbb{1}[Y \ne 0]].$$

*Let $\mathcal{F} \subset [0, 1]^{\mathcal{X}}$ be an L-Lipschitz class of predictors. Define the minimum joint error rate by $\lambda^* = \min_{f'}(\mathcal{R}_S(f') + \mathcal{R}_T(f'))$, then for all $f \in \mathcal{F}$,*
$$\mathcal{R}_T(f) \le \mathcal{R}_S(f) + 2L\, W_1(\mu_S^X, \mu_T^X) + \lambda^*.$$

*Proof.* Define random variable $\eta = \mathbb{1}[Y = 1]$, then we may rewrite
$$\mathcal{R}(f) = \mathbb{E}_{(X,Y)\sim\mu}[\eta - (2\eta - 1)f(X)].$$
Note that because $2\eta - 1 \in \{-1, +1\}$,
$$\mathcal{R}(f) - \mathcal{R}(f') = \mathbb{E}_{(X,Y)\sim\mu}[\eta - (2\eta - 1)f(X)] - \mathbb{E}_{(X,Y)\sim\mu}[\eta - (2\eta - 1)f'(X)]$$
$$= \mathbb{E}_{(X,Y)\sim\mu}[(2\eta - 1)(f'(X) - f(X))]$$
$$\le \mathbb{E}_{X\sim\mu^X}[|f'(X) - f(X)|].$$

On the other hand,

$$
\begin{aligned}
\mathbb{E}_{X\sim\mu^X}[|f(X)-f'(X)|] &= \mathbb{E}_{(X,Y)\sim\mu}[|(2\eta-1)(f(X)-f'(X))-\eta+\eta|] \\
&\le \mathbb{E}_{(X,Y)\sim\mu}[|(2\eta-1)f(X)-\eta|] + \mathbb{E}_{(X,Y)\sim\mu}[|-(2\eta-1)f'(X)+\eta|] \\
&= \mathbb{E}_{(X,Y)\sim\mu}[\eta-(2\eta-1)f(X)] + \mathbb{E}_{(X,Y)\sim\mu}[\eta-(2\eta-1)f'(X)] \\
&= \mathcal{R}(f') + \mathcal{R}(f).
\end{aligned}
$$

Then by Fact B.1, the fact that the operation of taking the absolute value is 1-Lipschitz, and Definition 3.2 of $W_1$ distance, for all $f, f' \in \mathcal{F}$,

$$
\begin{aligned}
\mathcal{R}_T(f) &= \mathcal{R}_S(f) + (\mathcal{R}_T(f)-\mathcal{R}_T(f')) - (\mathcal{R}_S(f)+\mathcal{R}_S(f')) + (\mathcal{R}_S(f')+\mathcal{R}_T(f')) \\
&\le \mathcal{R}_S(f) + \mathbb{E}_{X\sim\mu_T^X}[|f(X)-f'(X)|] - \mathbb{E}_{X\sim\mu_S^X}[|f(X)-f'(X)|] + (\mathcal{R}_S(f')+\mathcal{R}_T(f')) \\
&\le \mathcal{R}_S(f) + \sup_{q\in\mathrm{Lip}(2L)} (\mathbb{E}_{X\sim\mu_T^X}[q(X)] - \mathbb{E}_{X\sim\mu_S^X}[q(X)]) + (\mathcal{R}_S(f')+\mathcal{R}_T(f')) \\
&\le \mathcal{R}_S(f) + 2L\, W_1(\mu_S^X, \mu_T^X) + (\mathcal{R}_S(f')+\mathcal{R}_T(f')),
\end{aligned}
$$

and the result follows by taking the min over $f'$. $\qquad\square$

Next, we prove Theorem 4.7. The main idea is that under the setup and assumptions in Section 4, $\mathcal{R}_S$ and $\mathcal{R}_T$ can be written as the expectation of some Lipschitz function of $Z \sim \mu_S^Z$ and $\mu_T^Z$, respectively, so by Definition 3.2, their difference is upper bounded by a Lipschitz constant times $W_1(\mu_S^Z, \mu_T^Z)$.

Although omitted, we remark that Theorem 4.7 can be extended to the cutoff version of the ranking metric $u$ (e.g., NDCG@$p$) with a simple modification of the proof. Also, a finite sample generalization bound could be obtained using, e.g., Rademacher complexity, and additionally assuming Lipschitzness of the end-to-end scoring model [5, 51].

*Proof of Theorem 4.7.* Fix $g \in \mathcal{G}$, and consider $\mu_S^X$ for now (analogous results hold for $\mu_T^X$). Which by Assumption 4.6, when restricted to $\mathrm{supp}(\mu_S^X)$, has an $L_g$-Lipschitz inverse $g^{-1}$. Define function $\epsilon_{h,g} : \mathcal{Z} \to \mathbb{R}_{\ge 0}$ for a given $h : \mathcal{Z} \to \mathbb{R}^\ell$ as

$$
\begin{aligned}
\epsilon_{h,g}(z) &= \max_{r\in S_\ell} u(r, y_S \circ g^{-1}(z)) - \mathbb{E}_{R\sim p_{\exp(h(z))}}[u(R, y_S \circ g^{-1}(z))] \qquad (3) \\
&= \max_{r\in S_\ell} u(r, y_S \circ g^{-1}(z)) - \sum_{r\in S_\ell} u(r, y_S \circ g^{-1}(z)) \prod_{i=1}^{\ell} \frac{\exp(h(z)_{I(r)_i})}{\sum_{j=i}^{\ell} \exp(h(z)_{I(r)_j})},
\end{aligned}
$$

and note that $\mathcal{R}_S(h \circ g) = \mathbb{E}_{X\sim\mu_S^X}[\epsilon_{h,g}(g(X))] =: \mathbb{E}_{Z\sim\mu_S^Z}[\epsilon_{h,g}(z)]$. We show that $\epsilon_{h,g}$ is Lipschitz provided that $h$ is Lipschitz, from establishing the Lipschitzness of both terms in Eq. (3).

For the first term, because $u$ is $L_u$-Lipschitz w.r.t. $y_S \circ g^{-1}(z)$ and $y_S \circ g^{-1}(z)$ is in turn $L_y L_g$-Lipschitz w.r.t. $z$ by Assumptions 4.3 and 4.4, it follows that $z \mapsto u(r, y_S \circ g^{-1}(z))$ is $L_u L_y L_g$-Lipschitz w.r.t. $z$ for any fixed $r \in S_\ell$, and therefore so is the function $z \mapsto \max_{r\in S_\ell} u(r, y_S \circ g^{-1}(z))$ by Fact B.1.

For the second term, we show that it is Lipschitz (in Euclidean distance) w.r.t. both $y_S \circ g^{-1}(z) =: y$ and $h(z) =: s$. By Jensen's inequality and Fact B.1,

$$
\|\nabla_y \mathbb{E}_{R\sim p_{\exp(s)}}[u(R, y)]\|_2 = \|\mathbb{E}_{R\sim p_{\exp(s)}}[\nabla_y u(R, y)]\|_2 \le \mathbb{E}_{R\sim p_{\exp(s)}}[\|\nabla_y u(R, y)\|_2] \le L_u.
$$

And by the definition that $\nabla_x f(x) = [\frac{\partial}{\partial x_1} f(x), \cdots, \frac{\partial}{\partial x_d} f(x)]$ for any $f : \mathbb{R}^d \to \mathbb{R}$,

$$\|\nabla_s \mathbb{E}_{R \sim p_{\exp(s)}}[u(R, y)]\|_2$$

$$\leq \|\nabla_s \mathbb{E}_{R \sim p_{\exp(s)}}[u(R, y)]\|_1$$

$$= \sum_{m=1}^{\ell} \left| \sum_{r \in S_\ell} u(r, y) \left( \frac{\partial}{\partial s_{I(r)_m}} \prod_{i=1}^{\ell} \frac{\exp(s_{I(r)_i})}{\sum_{j=i}^{\ell} \exp(s_{I(r)_j})} \right) \right|$$

$$= \sum_{m=1}^{\ell} \left| \sum_{r \in S_\ell} u(r, y) \sum_{i=1}^{\ell} \left( \frac{\partial}{\partial s_{I(r)_m}} \frac{\exp(s_{I(r)_i})}{\sum_{j=i}^{\ell} \exp(s_{I(r)_j})} \right) \prod_{n \neq i} \frac{\exp(s_{I(r)_n})}{\sum_{j=n}^{\ell} \exp(s_{I(r)_j})} \right|$$

$$= \sum_{m=1}^{\ell} \sum_{r \in S_\ell} u(r, y) \left( \prod_{n=1}^{\ell} \frac{\exp(s_{I(r)_n})}{\sum_{j=n}^{\ell} \exp(s_{I(r)_j})} \right) \left| \sum_{i=1}^{m} \left( \mathbb{1}[m = i] - \frac{\exp(s_{I(r)_m})}{\sum_{j=i}^{\ell} \exp(s_{I(r)_j})} \right) \right|$$

$$\leq B \sum_{m=1}^{\ell} \sum_{r \in S_\ell} \left( \prod_{n=1}^{\ell} \frac{\exp(s_{I(r)_n})}{\sum_{j=n}^{\ell} \exp(s_{I(r)_j})} \right) \sum_{i=1}^{m} \left( \mathbb{1}[m = i] + \frac{\exp(s_{I(r)_m})}{\sum_{j=i}^{\ell} \exp(s_{I(r)_j})} \right)$$

$$= B \sum_{r \in S_\ell} \left( \prod_{n=1}^{\ell} \frac{\exp(s_{I(r)_n})}{\sum_{j=n}^{\ell} \exp(s_{I(r)_j})} \right) \sum_{i=1}^{\ell} \sum_{m=i}^{\ell} \left( \mathbb{1}[m = i] + \frac{\exp(s_{I(r)_m})}{\sum_{j=i}^{\ell} \exp(s_{I(r)_j})} \right)$$

$$= B \sum_{r \in S_\ell} \left( \prod_{n=1}^{\ell} \frac{\exp(s_{I(r)_n})}{\sum_{j=n}^{\ell} \exp(s_{I(r)_j})} \right) \left( \ell + \sum_{i=1}^{\ell} \frac{\sum_{m=i}^{\ell} \exp(s_{I(r)_m})}{\sum_{j=i}^{\ell} \exp(s_{I(r)_j})} \right)$$

$$= 2B\ell \sum_{r \in S_\ell} \left( \prod_{n=1}^{\ell} \frac{\exp(s_{I(r)_n})}{\sum_{j=n}^{\ell} \exp(s_{I(r)_j})} \right)$$

$$= 2B\ell,$$

where the second equality is due to the product rule, the third equality to $\frac{\mathrm{d}}{\mathrm{d}x_j} \mathrm{softmax}(x)_i = \mathrm{softmax}(x)_i (\mathbb{1}[i = j] - \mathrm{softmax}(x)_j)$, the last inequality to Assumption 4.3, and the last equality to identifying the pmf of the P-L model (Definition 4.1). Because $h \in \mathrm{Lip}(L_h)$ by Assumption 4.5, the two results above imply that the second term in Eq. (3) is Lipschitz: for all $z, z' \in \mathcal{Z}$,

$$\left| \mathbb{E}_{R \sim p_{\exp(h(z))}}[u(R, y_S \circ g^{-1}(z))] - \mathbb{E}_{R \sim p_{\exp(h(z'))}}[u(R, y_S \circ g^{-1}(z'))] \right|$$

$$\leq \left| \mathbb{E}_{R \sim p_{\exp(h(z))}}[u(R, y_S \circ g^{-1}(z))] - \mathbb{E}_{R \sim p_{\exp(h(z))}}[u(R, y_S \circ g^{-1}(z'))] \right|$$

$$+ \left| \mathbb{E}_{R \sim p_{\exp(h(z))}}[u(R, y_S \circ g^{-1}(z'))] - \mathbb{E}_{R \sim p_{\exp(h(z'))}}[u(R, y_S \circ g^{-1}(z'))] \right|$$

$$\leq L_u \|y_S \circ g^{-1}(z) - y_S \circ g^{-1}(z')\|_2 + 2B\ell \|h(z) - h(z')\|_2$$

$$\leq (L_u L_y L_g + 2B\ell L_h) \, d_{\mathcal{Z}}(z, z').$$

Combining the two parts, we have that $\epsilon_{h,g}$ is $2(L_u L_y L_g + B\ell L_h)$-Lipschitz in $z$.

Finally, by Fact B.1 and Definition 3.2, for all $g \in \mathcal{G}$ and $h, h' \in \mathcal{H}$,

$$\mathcal{R}_T(h \circ g) = \mathcal{R}_S(h \circ g) + (\mathcal{R}_T(h \circ g) - \mathcal{R}_T(h' \circ g)) - (\mathcal{R}_S(h \circ g) + \mathcal{R}_S(h' \circ g))$$
$$+ (\mathcal{R}_S(h' \circ g) + \mathcal{R}_T(h' \circ g))$$

$$\leq \mathcal{R}_S(h \circ g) + (\mathcal{R}_T(h \circ g) - \mathcal{R}_T(h' \circ g)) - (\mathcal{R}_S(h \circ g) - \mathcal{R}_S(h' \circ g))$$
$$+ (\mathcal{R}_S(h' \circ g) + \mathcal{R}_T(h' \circ g))$$

$$= \mathcal{R}_S(h \circ g) + \mathbb{E}_{Z \sim \mu_T^Z}[\epsilon_{h,g}(Z) - \epsilon_{h',g}(Z)] - \mathbb{E}_{Z \sim \mu_S^Z}[\epsilon_{h,g}(Z) - \epsilon_{h',g}(Z)]$$
$$+ (\mathcal{R}_S(h' \circ g) + \mathcal{R}_T(h' \circ g))$$

$$\leq \mathcal{R}_S(h \circ g) + \sup_{q \in \mathrm{Lip}(4(L_u L_y L_g + B\ell L_h))} (\mathbb{E}_{Z \sim \mu_T^Z}[q(Z)] - \mathbb{E}_{Z \sim \mu_S^Z}[q(Z)])$$
$$+ (\mathcal{R}_S(h' \circ g) + \mathcal{R}_T(h' \circ g))$$

$$\leq \mathcal{R}_S(h \circ g) + 4(L_u L_y L_g + B\ell L_h) \, W_1(\mu_S^Z, \mu_T^Z) + (\mathcal{R}_S(h' \circ g) + \mathcal{R}_T(h' \circ g)),$$

and the result follows by taking the min over $h'$. $\qquad \square$

Finally, we verify the Lipschitzness of RR and NDCG.

*Proof of Corollary 4.8.* Because $\mathrm{RR} \leq 1$ uniformly and $\|y - y'\|_2 \geq 1$ for all $y, y' \in \{0,1\}^\ell, y \neq y'$, so $y \mapsto \mathrm{RR}(r,y)$ is 1-Lipschitz. $\qquad\square$

*Proof of Corollary 4.9.* We show that NDCG is Lipschitz w.r.t. $y$ under the assumptions. Recall that

$$\mathrm{NDCG}(r,y) = \frac{\mathrm{DCG}(r,y)}{\mathrm{IDCG}(y)} = \left(\sum_{i=1}^\ell \frac{y_i}{\log(r_i^* + 1)}\right)^{-1} \sum_{i=1}^\ell \frac{y_i}{\log(r_i + 1)},$$

where $r^*$ is the descending order of $y$.

Note that IDCG is continuous and piecewise differentiable w.r.t. $y$ (each piece is associated with some $r' \in S_\ell$ and supported on $\{y : \arg\max_r \mathrm{DCG}(r,y) = r'\}$), then so is $\mathrm{IDCG}^{-1}$ on the set $\mathcal{Y}' := \{y : C^{-1} \leq \mathrm{IDCG}(y) \leq C\}$ (which is our assumption) by chain rule. Clearly, DCG is continuous and differentiable w.r.t. $y$, so NDCG is continuous and piecewise differentiable on $\mathcal{Y}'$ by product rule. By Fact B.1, to show that NDCG is Lipschitz, we just need to show that its gradient norm w.r.t. $y$ is bounded when evaluated on (the interior of) each piece.

Let $r \in S_\ell$ be arbitrary, and $y \in \mathcal{Y}'$ be s.t. it lies in the interior of the respective piece, namely, by denoting $r^*$ the descending order of $y$, the interior of the set $\{y' : \arg\max_r \mathrm{DCG}(r,y') = r^*\} = \{y' : y'_{I(r^*)_1} \geq y'_{I(r^*)_2} \geq \cdots \geq y'_{I(r^*)_\ell}\}$ (recall that $I(r)_i$ denotes the index of the item with rank $i$). Then, for all $k \in [\ell]$,

$$\left|\frac{\partial}{\partial y_k}\mathrm{NDCG}(r,y)\right| = \left|\mathrm{IDCG}(y)^{-1}\frac{\partial}{\partial y_k}\sum_{i=1}^\ell \frac{y_i}{\log(r_i+1)} - \mathrm{DCG}(r,y)\left(\frac{\partial}{\partial y_k}\sum_{i=1}^\ell \frac{y_i}{\log(r_i^*+1)}\right)^{-2}\right|$$

$$\leq \left|\mathrm{IDCG}(y)^{-1}\log(r_k+1)^{-1}\right| + \left|\mathrm{DCG}(r,y)\log(r_k^*+1)^2\right|$$

$$\leq \left|\mathrm{IDCG}(y)^{-1}\log(2)^{-1}\right| + \left|\mathrm{DCG}(r,y)\log(\ell+1)^2\right|$$

$$\leq C + C\log(\ell+1)^2,$$

hence $\|\nabla_y\mathrm{NDCG}(r,y)\|_2 \leq \sqrt{\ell}(C + C\log(\ell+1)^2) \leq \widetilde{O}(C\sqrt{\ell})$, and NDCG is $\widetilde{O}(C\sqrt{\ell})$-Lipschitz w.r.t. $y$. Here, $\widetilde{O}$ hides polylogarithmic terms. $\qquad\square$

# C    Additional Experiments and Details on Passage Reranking

We perform additional experiments for unsupervised domain adaptation on the passage reranking task considered in Section 5 in Appendix C.1, provide case studies on ListDA vs. zero-shot and QGen PL (Tables 8 and 9), hyperparameter settings in Appendix C.2, and details on the construction of training lists in Appendix C.3.

## C.1    Additional Results

**Pairwise Logistic Ranking Loss.**    Besides the listwise softmax cross-entropy ranking loss used in Section 5:

$$\ell(s,y) = -\sum_{i=1}^\ell y_i \log\left(\frac{\exp(s_i)}{\sum_{j=1}^\ell \exp(s_j)}\right),$$

we experiment with the pairwise logistic ranking loss [8]:

$$\ell(s,y) = -\sum_{i=1}^\ell \sum_{j=1}^\ell \mathbb{1}[y_i > y_j] \log\left(\frac{\exp(s_i)}{\exp(s_i) + \exp(s_j)}\right).$$

The results with using the pairwise logistic loss for training on the Robust04 dataset are provided in Table 2a, which are not better than using the softmax cross-entropy loss (cf. Table 1; see also [25]), hence further experiments with this loss are not pursued.

Table 2: Reranking performance of RankT5 on top 1000 BM25-retrieved passages; additional results extending Table 1.

(a) With pairwise logistic ranking loss in place of softmax cross-entropy on Robust04.

| Target domain | Method | MAP | MRR@10 | NDCG@5 | NDCG@10 | NDCG@20 |
|---|---|---|---|---|---|---|
| Robust04 | Zero-shot | 0.2656 | 0.7894 | 0.5671 | 0.5163 | 0.4729 |
| | QGen PL | 0.2776* | 0.7975 | 0.5576 | 0.5267 | 0.4892* |
| | ItemDA (MLP) | 0.2766* | 0.8021 | 0.5794 | 0.5355* | 0.4917* |
| | ListDA | 0.2893*†‡ | 0.8103 | 0.5935*†‡ | 0.5524*†‡ | 0.5044*†‡ |

Source domain is MS MARCO. Gain function in NDCG is the identity map. *Improves upon zero-shot with statistical significance ($p \leq 0.05$) under the two-tailed Student's $t$-test. †Improves upon QGen PL. ‡Improves upon ItemDA.

(b) With ItemDA with a transformer discriminator.

| Target domain | Method | MAP | MRR@10 | NDCG@5 | NDCG@10 | NDCG@20 |
|---|---|---|---|---|---|---|
| Robust04 | | 0.2848 | 0.8018 | 0.5851 | 0.5406 | 0.4982 |
| TREC-COVID | ItemDA (transformer) | 0.3088 | 0.8907 | 0.8287 | 0.8069 | 0.7691 |
| BioASQ | | 0.4739 | 0.6420 | 0.5256 | 0.5312 | 0.5567 |

Source domain is MS MARCO. Gain function in NDCG is the identity map.

(c) With ListDA + QGen PL method.

| Target domain | Method | MAP | MRR@10 | NDCG@5 | NDCG@10 | NDCG@20 |
|---|---|---|---|---|---|---|
| Robust04 | | 0.2851*† | 0.8039† | 0.5761† | 0.5386† | 0.4975† |
| TREC-COVID | ListDA + QGen PL | 0.3168 | 0.8950 | 0.8539 | 0.8292 | 0.7820 |
| BioASQ | | 0.6538*‡ | 0.5158 | 0.5547‡ | 0.5671*‡ | 0.5931*‡ |

Source domain is MS MARCO. Gain function in NDCG is the identity map. *Improves upon zero-shot with statistical significance ($p \leq 0.05$) under the two-tailed Student's $t$-test. †Improves upon QGen PL. ‡Improves upon ItemDA.

Table 3: Reranking performance of RankT5 on top 1000 BM25-retrieved passages on Signal-1M.

| Target domain | Method | MAP | MRR@10 | NDCG@5 | NDCG@10 | NDCG@20 |
|---|---|---|---|---|---|---|
| Signal-1M | BM25 | 0.1740 | 0.5765 | 0.3639 | 0.3215 | 0.2905 |
| | Zero-shot | 0.1511 | 0.4804 | 0.3068 | 0.2685 | 0.2410 |
| | QGen PL | 0.1541 | 0.5043 | 0.3238 | 0.2799 | 0.2497 |
| | ListDA | 0.1456 | 0.4629 | 0.3002 | 0.2602 | 0.2328 |
| | ListDA + QGen PL | 0.1549 | 0.5170 | 0.3261 | 0.2817 | 0.2505 |

Source domain is MS MARCO. Gain function in NDCG is the identity map.

As an implementation remark, in this set of experiments, we do not perform pairwise comparisons to obtain the predicted rank assignments during inference (where lists have length-1000) due to the high time complexity (the loss is aggregated pairwise during training, since we truncate training lists to length-31; see Appendix C.3). Whether or not the forward pass of the model involves pairwise computations does not matter to ListDA, which is applicable to any (pointwise, pairwise, or listwise) model as long as we can gather list-level representations; ListDA could be instantiated on pairwise models e.g. DuoT5 [44], although not pursued in this work.

**ItemDA with Transformer Discriminator.** To show that the improvements brought by ListDA over ItemDA is due to learning list-level invariant representations and not because of the use of the more expressive transformer as the discriminator vs. MLP, we experiment with a variant of ItemDA that uses the same transformer discriminator used in our ListDA runs. Although, we remark that whereas each input to the ListDA discriminator is a list of feature vectors as a length-$\ell$ sequence, each input to the ItemDA transformer discriminator is a single feature vector, or, a sequence of length-1. So, the attention mechanism of the transformer becomes pointless, and the ItemDA transformer discriminator collapses to an MLP (with skip connections and LayerNorm).

The results are presented in Table 2b (cf. Table 1). Across all datasets and metrics, no consistent improvement of ItemDA (transformer) over ItemDA (MLP) is observed.

**ListDA + QGen PL Method, and Signal-1M Dataset.** We experiment with supplementing ListDA with QGen pseudolabels by (uniformly) combining the training objectives of ListDA and QGen PL (**ListDA + QGen PL**). The results are presented in Table 2c. We also apply this method on the Signal-1M (RT) dataset [53], with results in Table 3. It is observed that reranking using neural rankers transferred from MS MARCO source domain does not perform better than BM25 (i.e., without any reranking) on Signal-1M, which is also noted in prior work [57, 32]. This does not mean that neural rerankers are worse than BM25, but could be that MS MARCO is not a good choice as the source domain for the target of Signal-1M, because of the arguably large domain shift between tweet retrieval and MS MARCO web search—qualitatively, it can be seen from Table 7 that the text styles and task semantics of the two domains are very different. Hence, the following discussions on Signal-1M results focus on comparing reranking results.

On Signal-1M, QGen PL improves upon the zero-shot baseline, but ListDA does not, which is likely due to the large domain shift between MS MARCO and Signal-1M that prevented ListDA from finding the correct source-target feature alignment without supervision. By incorporating QGen pseudolabels, ListDA performance improves with + QGen PL, which could have benefited from QGen q-d pairs acting as anchor points for ListDA to find the correct correspondence between source and target.

Overall, ListDA + QGen PL is the only method that consistently improves upon the zero-shot baseline on all four datasets, including Signal-1M, although it underperforms ListDA without QGen PL on the other three. Further improvements to this hybrid method may be possible with better strategies for balancing the training objectives of ListDA and QGen PL.

**Case Studies.** In Tables 8 and 9, we include examples where the ListDA top-1 reranked results are better than those of zero-shot and QGen PL, respectively, for qualitatively understanding the benefits and advantages of ListDA.

In zero-shot, the reranker is trained on the general domain MS MARCO dataset and applied on the target domains without adaptation. When the target domain differs from MS MARCO stylistically or consists of passages containing domain-specific words, as in the TREC-COVID and BioASQ datasets for biomedical retrieval, the zero-shot model, which has not seen the specialized language usages, may resort to keyword matching. Examples of such cases are presented in Table 8, where the top-1 passages reranked by the zero-shot model contain keywords from the query but are irrelevant.

In QGen PL, the reranker is trained by assuming the QGen synthesized query-document pairs to be relevant. However, as remarked in Section 5, because QGen is trained on MS MARCO and applied on the target domains in a zero-shot manner, the pseudolabels are not guaranteed to be valid, meaning that there could be false positives. This is observed in examples presented in Tables 7 and 9. One specific scenario where false positives hurt transfer performance is when the synthesized queries (of false positive documents) coincide with queries from the evaluation set, observed from the examples presented in Table 9 on TREC-COVID and BioASQ datasets. On the other hand, ListDA does not assume the pseudolabels in its training objective, so it does not suffer from these potential issues with QGen PL.

## C.2 Hyperparameter Settings

For BM25, we use the implementation of Anserini [69], set $k_1 = 0.82$ and $b = 0.68$ on MS MARCO source domain, and $k_1 = 0.9$ and $b = 0.4$ on all target domains without tuning. As in [57], titles are indexed as a separate field with equal weight as the document body, when available.

For the RankT5 reranking model, it is fine-tuned from the T5 v1.1 base checkpoint on a Dragonfish TPU with 8x8 topology for 100,000 steps with a batch size of 32 per domain (each training list contains 31 items). We tune the learning rate $\eta_{rank} \in \{$5e-5, 1e-4, 2e-4$\}$, and select the one with the best zero-shot performance to use on all adaptation methods, on each dataset (see Fig. 5 for zero-shot sweep results). We apply a learning rate schedule on $\eta_{rank}$ that decays (exponentially) by a factor of 0.7 every 5,000 steps. The concatenated query-document text inputs are truncated to 512 tokens.

For the domain discriminators, there are two hyperparameters: the strength of invariant feature learning $\lambda$, and the discriminator learning rate $\eta_{ad}$. We tune both by sweeping $\lambda \in \{0.01, 0.02\}$, and $\eta_{ad} \in \{10, 20, 40\} \times \eta_{rank}$ as multiples of the reranker learning rate. The tuned hyperparameter settings of $\eta_{rank}$, $\eta_{ad}$ and $\lambda$ used in our experiments are included in Table 4.

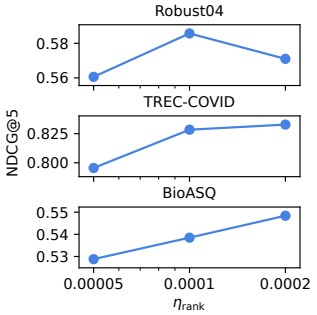
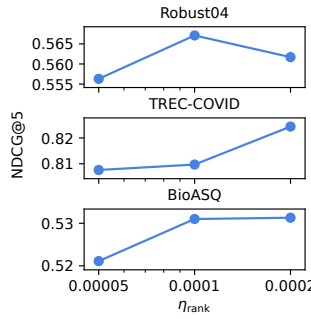

(a) With softmax cross-entropy ranking loss.  (b) With pairwise logistic ranking loss.

Figure 5: Zero-shot performance under different hyperparameter settings for $\eta_{\text{rank}}$.

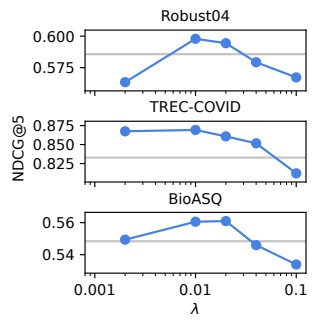
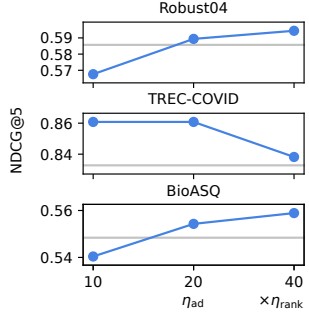

(a) Fix $\eta_{\text{ad}} = 0.004$ and sweep $\lambda$.  (b) Fix $\lambda = 0.02$ and sweep $\eta_{\text{ad}}$ (as multiples of $\eta_{\text{rank}}$).

Figure 6: ListDA performance under different hyperparameter settings for $\lambda$ and $\eta_{\text{ad}}$. Grey horizontal line is zero-shot performance with the best $\eta_{\text{rank}}$ setting.

The sweep results for ListDA are plotted in Fig. 6. It is observed that a balanced choice of $\lambda$ is needed to elicit the best performance from ListDA, and the same choice works fairly well across datasets. But for $\eta_{\text{ad}}$, each dataset prefers different settings, probably due to their distinct domain characteristics.

In terms of running time, the adaptation methods of ListDA, ItemDA, and QGen PL all have double the training time compared to zero-shot learning due to data loading. This is because in addition to source domain data, the adaptation methods require target domain (unlabeled) data. Among ListDA, ItemDA, and QGen PL, the training times are roughly the same, because the overhead of the domain discriminators is not significant.

## C.3 Training List Construction

Recall from the setup of ListDA on ranking problems that, the inputs are defined over lists and the invariant representations are learned at the list level. In other words, the ranking loss and the adversarial loss are to be computed on lists of ranking scores and feature vectors for each query (containing both relevant and irrelevant ones).

In our reranking setup, each list would be the top-$r$ passages retrieved by BM25 on a query, where we set $r = 1000$ in our evaluations. However, there are two complications. The first is that due to memory constraints, it is not feasible to feed all 1000 q-d pairs from each list through the RankT5 model in one pass during training (or time-consuming if via gradient accumulation). The second is that the source domain MS MARCO dataset only labels one relevant passage for each query, meaning that out of the 1000 passages retrieved by BM25 for each query, we may only know that (at most) one of them is relevant; the labels for the remaining 999 passages are unknown.

**Negative Sampling.** To address both issues, we truncate the list to length-31 for training and perform random sampling to gather irrelevant q-d pairs from the top 1000 BM25 retrieved passages.

Table 4: Hyperparameter settings of RankT5 model and domain discriminators for passage reranking experiments.

(a) With softmax cross-entropy ranking loss.

| Target domain | Method | $\eta_{\text{rank}}$ | $\eta_{\text{ad}}$ | $\lambda$ |
|---|---|---|---|---|
| Robust04 | Zero-shot | 1e-4 | - | - |
| | QGen PL | | - | - |
| | ItemDA (MLP) | | 1e-3 | 0.01 |
| | ItemDA (transformer) | | 2e-3 | 0.01 |
| | ListDA | | 4e-3 | 0.01 |
| | ListDA + QGen PL | | 1e-3 | 0.02 |
| TREC-COVID | Zero-shot | 2e-4 | - | - |
| | QGen PL | | - | - |
| | ItemDA (MLP) | | 8e-3 | 0.01 |
| | ItemDA (transformer) | | 2e-3 | 0.01 |
| | ListDA | | 4e-3 | 0.01 |
| | ListDA + QGen PL | | 4e-3 | 0.02 |
| BioASQ | Zero-shot | 2e-4 | - | - |
| | QGen PL | | - | - |
| | ItemDA (MLP) | | 4e-3 | 0.01 |
| | ItemDA (transformer) | | 2e-3 | 0.01 |
| | ListDA | | 8e-3 | 0.01 |
| | ListDA + QGen PL | | 4e-3 | 0.02 |
| Signal-1M | Zero-shot | 5e-5 | - | - |
| | QGen PL | | - | - |
| | ListDA | | 2e-3 | 0.01 |
| | ListDA + QGen PL | | 1e-3 | 0.02 |

(b) With pairwise logistic ranking loss.

| Target domain | Method | $\eta_{\text{rank}}$ | $\eta_{\text{ad}}$ | $\lambda$ |
|---|---|---|---|---|
| Robust04 | Zero-shot | 1e-4 | - | - |
| | QGen PL | | - | - |
| | ItemDA (MLP) | | 2e-3 | 0.01 |
| | ListDA | | 2e-3 | 0.01 |

On the MS MARCO source domain, given a query $q$ and the top 1000 passages retrieved by BM25, $d_1, \cdots, d_{1000}$, we yield a training list by bundling the one passage $d^*$ labeled as relevant with 30 randomly sampled passages $d_{N_1}, \cdots, d_{N_{30}}$ to be treated as irrelevant, i.e., $x = ([q, d^*], [q, d_{N_1}], \cdots, [q, d_{N_{30}}])$ and $y = (1, 0, \cdots, 0)$.

On the target domains, we follow the same procedure but with QGen synthesized queries. Given a QGen query $q$ and the top 1000 retrieved passages, the training list consists of the pseudolabeled passage $\hat{d}$ (i.e., the passage on which the query $q$ was synthesized) and 30 randomly sampled irrelevant passages, i.e., $x = ([q, \hat{d}], [q, d_{N_1}], \cdots, [q, d_{N_{30}}])$ and $y = (1, 0, \cdots, 0)$. Note that the pseudolabels $y$ on the target domain are used by QGen PL for training but discarded by ListDA.

**Reducing False Negatives on MS MARCO.** One potential problem arising from sampling is that the constructed lists may contain false negatives (i.e., relevant documents that are incorrectly treated as irrelevant). In fact, false negatives are prevalent in the MS MARCO dataset (e.g., due to duplicate passages). While these false negatives are mostly harmless to source domain supervised training because the effects are canceled out when there are positively labeled q-d pairs to which the false negative q-d pairs are similar, false negatives can affect invariant representation learning.

One way that it may negatively affect invariant representation learning is that the duplicates in the lists (which have identical feature vectors) will cause ListDA to collapse distinct passages on the target domain to the same feature vector for achieving alignment, which is an artifact that will cause information loss in the target domain feature representations.

To lower the chance of sampling false negatives on MS MARCO, we rerank BM25-retrieved passages using a ranker pre-trained on MS MARCO, and sample negatives from passages ranked at 300 or lower, as the duplicates and (unlabeled) relevant passages are concentrated at the top [47]. We apply this sampling method only when constructing source domain lists for feature alignment (namely, in

ListDA and ItemDA), and it only affects the computation of adversarial loss. This sampling method is not applicable to the target domains because we do not have reliable pre-trained rankers for them. It is also not used for constructing the lists for source domain supervised training (i.e., the computation of ranking loss), because it excludes "hard negatives" from the training lists and results in a weaker reranker, as observed in our preliminary experiments.

## D   Experiments on Yahoo! LETOR

We evaluate ListDA on the Yahoo! Learning to Rank Challenge v2.0 [11], and compare it to zero-shot and ItemDA (QGen PL is not applicable to tabular data). It is a web search ranking dataset with two subsets, "Set 1" and "Set 2", whose data originate from the US and a country in Asia, respectively; the training set sizes are 19,944 and 1,266, and the test set sizes are 6,983 and 3,798 (of which there are 47 and 17 lists of length-1). The dataset is tabular, and each item is represented by a 700-d vector with values in the range of $[0, 1]$. Among the 700 features, 415 are defined on both sets (shared), and the other 285 are defined on either Set 1 or 2 (disjoint); we hence write each item $x := [x_{\text{shared}}, x_{\text{disjoint}}]$ as a concatenation of the shared features and the disjoint ones, $x_{\text{shared}} \in \mathbb{R}^{415}, x_{\text{disjoint}} \in \mathbb{R}^{285}$.

We consider unsupervised domain adaptation from Set 1 to Set 2 (i.e., we discard the labels in Set 2). Our implementation uses PyTorch and the Hugging Face Transformers library [66].[7]

**Models.**   Our models have the same setup as that of the passage reranking experiments in Section 5, except that the RankT5 text ranking model is replaced by a three-hidden-layer MLP following [75], and we treat the list of 256-d outputs at the last hidden layer as feature representations:

$$g(x)_i =: v_i = \text{ReLU}\left( W_3 \begin{bmatrix} \text{ReLU}(W_2\text{ReLU}(W_1 x_{i,\text{shared}} + b_1) + b_2) \\ \text{ReLU}(W_2'\text{ReLU}(W_1' x_{i,\text{disjoint}} + b_1') + b_2') \end{bmatrix} + b_3 \right),$$

$$h(x)_i =: s_i = W_4 v_i + b_4,$$

where $W_1 \in \mathbb{R}^{1024 \times 415}$, $W_1' \in \mathbb{R}^{1024 \times 285}$, $W_2, W_2' \in \mathbb{R}^{256 \times 1024}$, $W_3 \in \mathbb{R}^{256 \times 512}$, and $W_4 \in \mathbb{R}^{1 \times 256}$, and are randomly initialized. Note that we use separate input layers ($W_1, W_1'$) and first hidden layers ($W_2, W_2'$) for the shared and disjoint features, then concatenate the hidden representations. This tweak improves performance of the zero-shot baseline.

For ItemDA, we use the same ensemble of five three-layer MLPs as in Section 5. For the ListDA discriminator, an ensemble of five transformers is used, where each transformer is a stack of three T5 encoder blocks, with 4 attention heads (`num_heads`), size-32 key, query and value projections per attention head (`d_kv`), and size-1024 intermediate feedforward layers (`d_ff`).

**Results.**   The results are presented in Table 5, which are ensembles of 10 separately trained models to reduce the variance from the randomness in the initialization and training process, due the small dataset size. Yahoo! LETOR is annotated with 5-level relevancy, and the scores are binarized for MAP and MRR metrics by mapping 0 (bad) and 1 (fair) to negative, and 2 (good), 3 (excellent), 4 (perfect) to positive. Thanks to the availability of labeled data on Set 2, we also include **supervised** results from training on Set 1 and 2 to serve as an upper bound for unsupervised domain adaptation.

ListDA again outperforms ItemDA for unsupervised domain adaptation, collaborating the discussions in the main sections of the paper. Although, here, their performance gap is smaller compared to passage reranking results, which we suspect is because the contextual (i.e., query) information for defining the list structure of the data is too weak or discarded when the numerical features are generated (e.g., the numerical features could represent the difference between query and document, hence may not retain information about the original query).

**Hyperparameters.**   The model is trained from scratch on an NVIDIA A6000 GPU for 10,000 steps with a batch size of 32 per domain (length of training lists varies from one to 140 items). We apply a learning rate schedule on $\eta_{\text{rank}}$ that decays (exponentially) by a factor of 0.7 every 500 steps.

We tune the hyperparameters by performing grid search over: $\eta_{\text{rank}} \in \{$1e-5, 2e-5, 4e-5, 8e-5, 1e-4, 2e-4, 4e-4, 8e-4, 1e-3$\}$, $\eta_{\text{ad}} \in \{0.2, 0.4, 0.8, 1, 2, 4, 8, 10\} \times \eta_{\text{rank}}$ as multiples of the ranker learning rate, and $\lambda \in \{0.08, 0.1, 0.2, 0.4, 0.8, 1\}$. The tuned settings are included in Table 6.

---

[7]Our code for this set of experiments is available in the supplementary material, and the data can be requested at https://webscope.sandbox.yahoo.com/catalog.php?datatype=c.

Table 5: Ranking performance of MLP ranker on Yahoo! LETOR (Set 2).

| Target domain | Method | MAP | MRR@10 | NDCG@5 | NDCG@10 | NDCG@20 |
|---|---|---|---|---|---|---|
| Yahoo! LETOR (Set 2) | Zero-shot | 0.5464 | 0.6796 | 0.7466 | 0.7778 | 0.8308 |
| | Supervised | 0.5714 | 0.7045 | 0.7760 | 0.8012 | 0.8479 |
| | ItemDA | 0.5604* | 0.6984* | 0.7563* | 0.7822 | 0.8350* |
| | ListDA | **0.5633***  | **0.7026***  | **0.7599***  | **0.7845***  | **0.8355***  |

Results are from an ensemble of five scoring models. Bold indicates the best unsupervised result. Source domain is Yahoo! LETOR (Set 1). Gain function in NDCG is the identity map. *Improves upon zero-shot with statistical significance ($p \leq 0.05$) under the two-tailed Student's $t$-test. ‡Improves upon ItemDA. Significance tests are not performed on supervised results.

Table 6: Hyperparameter settings of MLP ranker and domain discriminators for Yahoo! LETOR experiments.

| Target domain | Method | $\eta_{\text{rank}}$ | $\eta_{\text{ad}}$ | $\lambda$ |
|---|---|---|---|---|
| Yahoo! LETOR (Set 2) | Zero-shot | 8e-4 | - | - |
| | Supervised | 4e-5 | - | - |
| | ItemDA | 8e-4 | 1.6e-3 | 0.4 |
| | ListDA | 8e-4 | 1.6e-3 | 0.8 |

Table 7: Examples of relevant q-d pairs from the test set and pairs synthesized by QGen on domains considered in Section 5 experiments. Text truncation or omission are indicated by "[...]".

| Dataset | Relevant q-d pairs | QGen q-d pairs |
|---|---|---|
| MS MARCO | D: What is cartography? A. the science of map-making B. the science of shipbuilding C. the science of charting direction on a ship D. the science of measuring distances on the ocean. Cartography is the science of map making A. 

 Q: what is the science of mapmaking called | - |
| | D: The flu shot also contains the following ingredients: sodium phosphate & buffered isotonic sodium chloride solution, formaldehyde, octylphenol ethoxylate, and gelatin, according to the FDA. 

 Q: what's in the flu shot | |
| TREC-COVID | D: An Evidence Based Perspective on mRNA-SARS-CoV-2 Vaccine Development. [...] The production of mRNA-based vaccines is a promising recent development in the production of vaccines. However, there remain significant challenges in the development [...] 

 Q: what is known about an mRNA vaccine for the SARS-CoV-2 virus? | D: Impact of arterial load on the agreement between pulse pressure analysis and esophageal Doppler. INTRODUCTION. The reliability of pulse pressure analysis to estimate cardiac output is known to be affected by arterial load changes. [...] 

 QGen: what is arterial load for pulse pressure analysis |
| | D: The possible pathophysiology mechanism of cytokine storm in elderly adults with COVID-19 infection: the contribution of "inflame-aging". PURPOSE: Novel Coronavirus disease 2019 (COVID-19), is an acute respiratory distress syndrome (ARDS), [...] 

 Q: What is the mechanism of cytokine storm syndrome on the COVID-19? | D: Opportunity Costs Pacifism. If the resources used to wage wars could be spent elsewhere and save more lives, does this mean that wars are unjustified? This article considers this question, which has been largely overlooked by Just War Theorists and pacifists. It focuses on whether the opportunity costs of war [...] 

 QGen: opportunity cost pacifism |
| BioASQ | D: The role of extended-release amantadine for the treatment of dyskinesia in Parkinson's disease patients. [...] Extended-release amantadine (amantadine ER) is the first approved medication for the treatment of dyskinesia. When it is given at bedtime, it [...] 

 Q: Is amantadine ER the first approved treatment for akinesia? | D: Subluxation of the femoral head in coxa plana. Twenty-two patients who had severe coxa plana had closed reduction for lateral subluxation of the femoral head, [...] The average age when the patients were first seen was eight years and six months. [...] 

 QGen: average age of femoral subluxation |
| | D: [...] We investigated the health-related quality of life (HRQoL) of long-term prostate cancer patients who received leuprorelin acetate in microcapsules (LAM) for androgen-deprivation therapy (ADT). [...] 

 Q: Can leuprorelin acetate be used as androgen deprivation therapy? | D: [...] a comparison of proxy assessment and patient self-rating using the disease-specific Huntington's disease health-related quality of life questionnaire (HDQoL). [...] Specific Scales of the HDQoL. On the Specific Hopes and Worries Scale, proxies on average rated HrQoL as better than patients' [...] 

 QGen: which scale is used for proxy assessment of hrqol |
| Signal-1M | D: BJP terms party MP R.K Singh's allegation that money has changed hands for tickets in #BiharPolls as baseless. 

 Q: Party MP calls BJP 'Baura Jayewala Party' | D: Black lives matter: thoughts from the delivery ward in St. Louis: #mustread 

 QGen: where is black lives matter? |
| | D: Kerry: US plans military talks with Russia over Syria 

 Q: Kerry: US plans military talks with Russia over Syria | D: RETWEET if "Brenda's Got A Baby" is one of your favorite @2Pac songs. #RIP2Pac 

 QGen: brenda got a baby pac |

Table 8: Examples where ListDA top-1 reranked result is better than zero-shot. Text truncation or omission are indicated by "[...]".

| Dataset | Zero-shot top-1 passage (that are irrelevant) | ListDA top-1 passage (that are relevant) |
|---|---|---|
| Robust04 | Q: Find information on prostate cancer detection and treatment. | |
| | D: [...] FIRST PATIENT UNDERGOES GENE INSERTION IN CANCER TREATMENT [...] This first round of gene transfer experiments, in which a gene was inserted into a patient's white blood cells, is not expected to directly benefit an individual patient. Instead, the inserted gene is being used to track the movement in the body of the cancer-fighting white blood cells. [...] Inserting human genes to repair defects may one day help with a host of inherited disorders, [...] | D: [...] Little knowledge goes a long way - Cancer of the prostate need not be a killer / Health Check. Earlier this year, 13-year-old [...] died from cancer of the bladder and prostate. His death is a grim reminder that no male should consider himself immune from waterworks trouble. The prostate, a gland about the size and shape of a chestnut, lies deep in the pelvis just below the bladder. Because it surrounds the urethra, it has the potential to block the flow of urine completely. [...] |
| TREC-COVID | Q: What are the longer-term complications of those who recover from COVID-19? | |
| | D: [...] Our previous experience with members of the same corona virus family (SARS and MERS) which have caused two major epidemics in the past albeit of much lower magnitude, has taught us that the harmful effect of such outbreaks are not limited to acute complications alone. Long term cardiopulmonary, glucometabolic and neuropsychiatric complications have been documented following these infections. [...] | D: Up to 20-30% of patients hospitalized with coronavirus disease (COVID-19) have evidence of myocardial involvement. Acute cardiac injury in patients hospitalized with COVID-19 is associated with higher morbidity and mortality. There are no data on how acute treatment for COVID-19 may affect convalescent phase or long-term cardiac recovery and function. Myocarditis from other viral pathogens can evolve into overt or subclinical myocardial dysfunction, [...] |
| BioASQ | Q: What is the interaction between WAPL and PDS5 proteins? | |
| | D: Pds5 and Wpl1 act as anti-establishment factors preventing sister-chromatid cohesion until counteracted in S-phase by the cohesin acetyl-transferase Eso1. [...] Here, we show that Pds5 is essential for cohesin acetylation by Eso1 and ensures the maintenance of cohesion by promoting a stable cohesin interaction with replicated chromosomes. The latter requires Eso1 only in the presence of Wapl, indicating that cohesin stabilization relies on Eso1 only to neutralize the anti-establishment activity. [...] | D: [...] Here, we show that cohesin suppresses compartments but is required for TADs and loops, that CTCF defines their boundaries, and that the cohesin unloading factor WAPL and its PDS5 binding partners control the length of loops. In the absence of WAPL and PDS5 proteins, cohesin forms extended loops, presumably by passing CTCF sites, accumulates in axial chromosomal positions (vermicelli), and condenses chromosomes. [...] |

Table 9: Examples where ListDA top-1 reranked result is better than QGen PL, along with the QGen synthesized query for the top-1 passage. Text truncation or omission are indicated by "[...]".

| Dataset | QGen PL top-1 passage (that are irrelevant) | ListDA top-1 passage (that are relevant) |
|---|---|---|
| Robust04 | Q: Identify outbreaks of Legionnaires' disease. | |
| | D: [...] 3. Care of Patients with Tracheostomy 4. Suctioning of Respiratory Tract Secretions III. Modifying Host Risk for Infection A. Precautions for Prevention of Endogenous Pneumonia 1. Prevention of Aspiration 2. Prevention of Gastric Colonization B. Prevention of Postoperative Pneumonia C. Other Prophylactic Procedures for Pneumonia 1. Vaccination of Patients 2. Systemic Antimicrobial Prophylaxis 3. Use of Rotating "Kinetic" Beds Prevention and Control of Legionnaires' Disease [...] 
 QGen: what kind of precautions are used to prevent pneumonia | D: [...] LEGIONNAIRE'S DISEASE STRIKES 16 AT REUNION IN COLORADO; 3 DIE. An outbreak of legionnaire's disease at a 50th high school reunion was blamed Thursday for the deaths of three elderly celebrants and the pneumonia-like illness of 13 others. State health officials contacted 250 other people from 21 states who attended the Lamar High School reunion for the classes of 1937 through 1941 but found no new cases, Dr. Ellen Mangione, a Colorado Department of Health epidemiologist, said. [...] |
| TREC-COVID | Q: what drugs have been active against SARS-CoV or SARS-CoV-2 in animal studies? | |
| | D: Different treatments are currently used for clinical management of SARS-CoV-2 infection, but little is known about their efficacy yet. Here we present ongoing results to compare currently available drugs for a variety of diseases to find out if they counteract SARS-CoV-2-induced cytopathic effect in vitro. [...] We will provide results as soon as they become available, [...] 
 QGen: what is the treatment for sars | D: [...] the antiviral efficacies of lopinavir-ritonavir, hydroxychloroquine sulfate, and emtricitabine-tenofovir for SARS-CoV-2 infection were assessed in the ferret infection model. [...] all antiviral drugs tested marginally reduced the overall clinical scores of infected ferrets but did not significantly affect in vivo virus titers. Despite the potential discrepancy of drug efficacies between animals and humans, these preclinical ferret data should be highly informative to future therapeutic treatment of COVID-19 patients. |
| BioASQ | Q: What is the function of the Spt6 gene in yeast? | |
| | D: As a means to study surface proteins involved in the yeast to hypha transition, human monoclonal antibody fragments (single-chain variable fragments, scFv) have been generated that bind to antigens expressed on the surface of Candida albicans yeast and/or hyphae. [...] To assess C. albicans SPT6 function, expression of the C. albicans gene was induced in a defined S. cerevisiaespt6 mutant. Partial complementation was seen, confirming that the C. albicans and S. cerevisiae genes are functionally related in these species. 
 QGen: what is the function of spt6 gene in candida albicans | D: Spt6 is a highly conserved histone chaperone that interacts directly with both RNA polymerase II and histones to regulate gene expression. [...] Our results demonstrate dramatic changes to transcription and chromatin structure in the mutant, including elevated antisense transcripts at >70% of all genes and general loss of the +1 nucleosome. Furthermore, Spt6 is required for marks associated with active transcription, including trimethylation of histone H3 on lysine 4, previously observed in humans but not Saccharomyces cerevisiae, and lysine 36. [...] |

