# OpenReview forum: "Learning List-Level Domain-Invariant Representations for Ranking"
_NeurIPS.cc/2023/Conference — NeurIPS 2023 spotlight_

### Official Review · Reviewer_nxwQ · 2023-06-29

**Soundness:** 3 good
**Presentation:** 3 good
**Contribution:** 3 good
**Rating:** 7
**Confidence:** 4

**Summary:**

The authors propose a new domain-invariant ranking method which enforces list-wise feature invariance as opposed to item-wise feature invariance. The authors prove a novel domain adaptation bound specifically for the ranking problem. Authors show strong empirical improvement over item-wise invariance methods.

**Strengths:**

(1) The theorem is new and not a trivial extension of the existing domain adaptation generalization bound from Ben-David (which is for classification). In particular, some new techniques are required to analyze the ranking problem.

(2) The method is novel and results are good.

**Weaknesses:**

(1) My main concern is with the empirical improvement between ItemDA and ListDA. I remain unconvinced that this improvement is coming from the list-wise invariance and not from the improvement in discriminator architecture. The authors mention that ItemDA uses a three-layer MLP, while ListDA uses a stack of three transformer blocks. is there anyway to compare the two methods using the same discriminator architecture?

(2) While the novel generalization bound and algorithm are both novel. The theory doesn't explain why ListDA is better than ItemDA. Personally, I do not see this as an issue.

**Questions:**

I find the statement in line 141 confusing (even after looking at Appendix A.2). I would think that item-level invariance implies list-level invariance. For example, consider the case where $g$ was trained to output a domain-invariant feature representation of query-document pairs (at the item level). The ranker $h$ operates on these domain invariant features to form a list. How could the resulting list be different between the domains, if the ranker is constant and the output of $g$ is domain-invariant? Perhaps it would be helpful to elaborate on this point further in the paper.

---

> ### Author Rebuttal · Authors · 2023-08-09
>
> We thank the reviewer for the time, comments, and support! We hope that the following addresses the concerns:
>
> - **“the statement in line 141 [is] confusing”**
>
> 	Thank you for the question—this was also a conceptual hurdle for us when we worked on this paper (please kindly find the new figure attached to the global response that illustrates the difference between list-level and item-level alignment).
>
> 	In ranking problems (for information retrieval), the list is defined to be the q-d pairs with the same query q. Although in our experiment setup, $g$ computes feature vector (in $\mathbb R^k$) for each q-d pair in isolation, it does not mean that the list structure is lost: the list feature representation is the set of features $\\{g(q,d_1),\cdots, z_(q,d_\ell)\\}$ computed on q-d pairs with the same q.
>
> 	This means that the list-level feature distribution $\mu_Z$ is defined over the set of features vectors computed on q-d pairs with the same q (the support of this distribution is the space of sets that contain $\ell$ length-$k$ vectors). This is in contrast to the item-level feature distribution $\mu_V$, which is defined over the feature vectors, ignoring the list structure (the support is $\mathbb R^k$, the space of length-$k$ vectors).
>
> 	Therefore, even if item-level distributions are aligned, after imposing the list structure on the items based on whether they have the same q (each list contains not one but $\ell$ items), the resulting list-level distributions are no longer necessarily aligned. Or, from a different perspective, there are strictly more ways of learning representations that are aligned at the item-level than there are at the list-level, and this is what we intended to demonstrate with the example in Appendix A.2.
>
> - **“The theory doesn't explain why ListDA is better than ItemDA”**
>
> 	While we did not prove a separation, we discussed (in the first point above) and showed that item-level alignment does not imply list-level alignment, which is required by the generalization bound for domain transfer measured in terms of ranking metrics. This suggests that ItemDA, which discards the list structure, is not sufficient for domain alignment on ranking problems, and ListDA is the more appropriate choice.

---

> ### Comment · Reviewer_nxwQ · 2023-08-15
> **response to rebuttal**
>
> I thank the reviewers for the rebuttal and experiments. I am satisfied.
>
> My only concern regarding the discriminator architecture was addressed in the rebuttal experiments. They show that using the transformer architecture on the baseline provides no improvement.
>
> I thank the authors for answering my conceptual question regarding item-level alignment and list-level alignment. The figure in the attachment is especially helpful.

---

### Official Review · Reviewer_D69g · 2023-07-05

**Soundness:** 3 good
**Presentation:** 3 good
**Contribution:** 3 good
**Rating:** 6
**Confidence:** 4

**Summary:**

This paper focuses on domain adaptation in the context of ranking problems. The authors propose a novel approach called list-level alignment to learn domain-invariant representations for ranking tasks. They demonstrate the benefits of this approach through theoretical analysis and empirical experiments.

Existing domain adaptation methods for ranking, particularly invariant representation learning, have been sporadically used and lack theoretical guarantees.

Given the limitations of previous approaches, the authors aim to address the conceptual discrepancy between item-level alignment and the intrinsic list structure of ranking problems. They propose list-level alignment as a solution to learn higher-level invariant representations on rank lists. This approach not only provides a theoretical foundation for domain adaptation in ranking but also achieves better empirical performance.

**Strengths:**

I think this problem setting is interesting and practical. I have the following views on this paper's strengths part.

From the perspective of problem and method, this paper addresses the challenge of training ranking models in domains with limited annotated data. It introduces domain adaptation as a transfer learning technique and highlights the need for invariant representation learning in the ranking literature. The authors propose a new approach called \list-level alignment\ that preserves the list structure of ranking problems and achieves higher-level invariant representations on the lists. The benefits of this approach include the establishment of the first domain adaptation generalization bound for ranking and better empirical performance on unsupervised domain adaptation tasks.

Experimentally, the authors conducted experiments on unsupervised domain adaptation tasks to evaluate the performance of their proposed approach, list-level alignment. The proposed approach achieved better empirical performance on unsupervised domain adaptation tasks compared to existing methods. The approach also established the first domain adaptation generalization bound for ranking, providing theoretical guarantees for its implementation.

**Weaknesses:**

I have the following views on this paper's weakness part.

- The paper acknowledges that existing domain adaptation methods for ranking lack theoretical guarantees, and it claims to have theoretical foundations for its proposed approach of list-level alignment. It's already a good contribution. However, the contribution of this paper in the theoretical aspect may be considered weak, since the contribution of this paper can be seen as an extension of existing domain adaptation theorems[1] applied to the ranking problem. Correct me if I misunderstand it.

- One curious aspect here is that while the paper discusses domain adaptation (DA) in the ranking problem, the experiments are conducted on reranking. In the context of reranking, which involves reordering a list of candidate documents generated by an initial retrieval model in response to a search query. However, it does not directly address the fundamental ranking problem. This limitation of the paper should be further discussed to provide clarity.

[1] Ben-David, Shai, et al. "Analysis of representations for domain adaptation." Advances in neural information processing systems 19 (2006).

**Questions:**

In my first point at my weakness part, if I misunderstand the theoretical contributions, the author could correct me.

**Limitations:**

Yes, it's properly stated in the paper.

---

> ### Author Rebuttal · Authors · 2023-08-09
>
> We thank the reviewer for the time, comments, and support! We hope that the following addresses the concerns:
>
> - **“the theoretical aspect may be considered weak… can be seen as an extension of existing domain adaptation theorems”**
>
> 	The result established in [1] is in terms of accuracy for binary classification problems, not ranking, and we point out that their proof technique cannot be extended to get a bound in terms of ranking metrics (e.g., NDCG, MRR) for ranking problems, due to technical difficulties that arise from the problem setup for ranking. For instance, the tightness of the bound in [1] depends on the uniqueness of the optimal Bayes classifier, but the optimal ranker/scorer on ranking problems are generally not unique, as discussed in Section 4.
>
> 	In addition, our bound can only be obtained after recognizing the need for list-level representation alignment, rather than item-level alignment as done in prior work on (ranking) domain adaptation. This necessary conceptual leap—which enabled both the theoretical result and the empirical improvements—is also our main theoretical contribution.
>
> - **“the experiments are conducted on reranking… does not directly address the fundamental ranking problem”**
>
> 	In Appendix D, we performed experiments on the Yahoo! Learning to Rank (LETOR) Challenge v2.0 dataset, a web search ranking task on numerical data, where the lists to be ranked are directly defined by the dataset (as opposed to obtained from a retrieval stage), hence the setting is a “fundamental ranking problem”. The results are consistent with those from the passage reranking experiments and also support our findings in the main sections.
>
> 	Nevertheless, our theory applies to any data/problem setting that involves lists of items ($x$) and scores ($y$) with the metrics being ranking metrics (e.g., NDCG, MRR), and this includes both passage reranking and Yahoo! LETOR. The only difference between them, in our opinion, is how the raw data is stored (the former as a corpus of queries and documents; the latter directly stores the lists) and how the lists are formed (the former uses a retriever to form the lists, i.e., the retriever, which we do not adapt, is treated as part of the problem setup; the latter as-is).

---

### Official Review · Reviewer_sWYy · 2023-07-07

**Soundness:** 2 fair
**Presentation:** 3 good
**Contribution:** 2 fair
**Rating:** 5
**Confidence:** 3

**Summary:**

This paper proposes to learn domain-invariant representations for ranking. In contrast to prior works that typically consider item-level alignment, they introduce the concept of list-level alignment to learn higher-level invariant representations on the lists. The domain adaptation generalization bound is explicitly established and extensive experiments on benchmark datasets reveal the effectiveness of the proposed approach.

Post rebuttal: my major concerns have been adequately addressed, and thus I've decided to change my score from 4 to 5.

**Strengths:**

- Domain adaptation for ranking problems is significant but receives scant attention.

- The paper in general is well-motivated and presents major challenges that need to be overcome to address DA for ranking. A new paradigm termed list-level alignment is proposed.

- They evaluate and ablate the method in multiple benchmark datasets.

**Weaknesses:**

- The proposed alignment relies on the traditional domain adversarial training, offering no significant advancements. What are the differences between the proposed alignment and prior adversarial alignment, and why the proposed list-level alignment is preferable in the context of ranking problems? The primary distinctions between list-level alignment and other paradigms are not well-articulated.

- According to the reported results, the proposed ListDA shows marginal improvements over baseline methods in terms of MAP.

- The connection between the theory and the proposed method is not sufficiently clear and precise, leaving readers uncertain about their relationship.

**Questions:**

Please refer to the weakness.

---

> ### Author Rebuttal · Authors · 2023-08-09
>
> We thank the reviewer for the time, comments, and support! We hope that the following addresses the concerns:
>
> - **“The proposed alignment relies on the traditional domain adversarial training, offering no significant advancements. What are the differences between the proposed alignment and prior adversarial alignment”**
>
> 	The key distinction between adversarial training for our list-level invariance and prior work’s item-level invariance is the distribution we try to align. For item-level invariance (e.g., ItemDA), we try to align the distribution of item-level representations, while for list-level invariance, we treat the entire list as a whole, derive a list-level representation, and aim to align the distribution of the list-level representation (please also kindly find the figure in our global response).
>
> 	Another distinction lies in the choice of the discriminator (lines 130–140). ListDA requires list-level invariance, which means that the input of the discriminator is list-like. For modeling such data, we used a transformer model architecture for our ListDA discriminator; specifically, the transformer does not have positional encoding, so it is compatible with the permutation-invariance property of ranking data. In contrast, there is more freedom in the choice of discriminator for item-level domain alignment, and most prior work uses MLP.
>
> 	In short, our contribution is to provide an appropriate alignment objective for ranking problems—with rigorous justification (Theorem 4.6), and our focus is not on the optimization side (we note that besides adversarial training, invariant representations can be learned using other methods, incl. optimal transports [1, 2]). We adapted adversarial training for our use case because of its popularity in the literature.
>
> - **“why the proposed list-level alignment is preferable in the context of ranking problems”**
>
> 	Our generalization bounds in Corollaries 4.7–4.8 (derived from Theorem 4.6) state that, if the representations are aligned at the list-level, then target domain performance—in terms of ranking metrics (e.g., NDCG, MRR)—can be lower bounded by source domain performance. But we are not aware of any bound for ranking via item-level invariance; moreover, no such bound has been established for ranking in prior work, because the majority of them studied classification problems (which use accuracy as the metric).
>
> 	This means that item-level invariance does not have performance guarantees, but list-level invariance does so, and is hence more appropriate on ranking problems. This is verified in our experiments, where ListDA always outperforms ItemDA; in particular, because of the lack of guarantee, the latter can sometimes result in negative transfer. Another intuitive reason is that ItemDA does not take into account the list structure inherent in the data of ranking problems, and ListDA does; the distinction between list-level and item-level invariance is discussed on lines 141–149 (please also kindly find the new figure attached to the global response): list-level invariance is a stricter requirement than item-level invariance, suggesting that ItemDA is not sufficient for ranking problems.
>
> - **“The connection between the theory and the proposed method is not sufficiently clear and precise”**
>
> 	Corollaries 4.7–4.8 (derived from Theorem 4.6) hold when the representations are aligned at the list-level, and on the other hand, our ListDA method is proposed to learn list-level domain invariant representations.
>
> 	So the bounds provide guarantees for the method, and the method is aimed at maximizing the bounds. The two are connected by the concept of list-level invariance. We will elaborate upon the existing discussion on lines 203–209 and emphasize this connection.
>
> - **“ListDA shows marginal improvements over baseline methods in terms of MAP”**
>
> 	The ListDA method consistently achieves significantly better mean average precision (MAP) than ItemDA and zero-shot baselines on all three datasets. While the improvement over QGen PL may seem marginal, it is important to note that MAP is just one of several evaluation metrics used in our experiments, and ListDA still outperforms baseline methods on multiple evaluation metrics (including NDCG@k and MRR@10) across different datasets. We believe that the combination of these results supports the effectiveness of ListDA.
>
> 	Furthermore, the contribution of this paper extends beyond the proposed ListDA method. The list-level alignment framework provides a foundation for future research on domain adaptation in ranking tasks (e.g., more sophisticated discriminator structures).
>
> [1] Chen et al. A Gradual, Semi-Discrete Approach to Generative Network Training via Explicit Wasserstein Minimization. ICML 2019.
> [2] Zhou et al. Iterative Alignment Flows. AISTATS 2022.

---

> > ### Comment · Area_Chair_3ci7 · 2023-08-19
> >
> > I acknowledge the author's rebuttal and I am encouraging reviewers to comment on your reply.
> >
> > Best,
> > AC

---

### Official Review · Reviewer_3wjd · 2023-07-11

**Soundness:** 3 good
**Presentation:** 3 good
**Contribution:** 3 good
**Rating:** 7
**Confidence:** 4

**Summary:**

This paper proposes a novel approach for domain adaptation on list ranking problems. In significant contrast with previous works, they adopt aligning list-level representations directly instead of item-level representations, and argue for its relevance in improving on list-level metrics pertinent to the problem of ranking. They also devise a theoretical bound for the target error in terms of source error, a joint hypothesis error and the list-level domain divergence, under suitable assumptions. Results on passage re-ranking task shows their strength in practical applications.

**Strengths:**

- The paper is very well written, easy to follow and extremely well presented.

- The idea of extending UDA approaches to list-level re-ranking problems is interesting and highly relevant. The ideas discussed in the approach might be relevant to problems beyond ranking (like object detection in computer vision which also uses unique evaluation metrics).

- The empirical results on the passage re-ranking task is strong.

- The simple of idea of using a transformer discriminator to allow permutation in-variance is impressive.

**Weaknesses:**

- Few choices, like the use of transformer discriminator instead of MLP, wasn't evaluated or ablated empirically, although it makes sense intuitively.

- The authors might also consider showing results on diverse ranking tasks beyond passage re-ranking, possibly in tasks like fine-grained computer vision as in [1], although I agree that this is much beyond the scope of this work.

[1] Wang, Xinshao, et al. "Ranked list loss for deep metric learning." Proceedings of the IEEE/CVF conference on computer vision and pattern recognition. 2019.

**Questions:**

Is there a typo in L98? Since both $\mathcal{L}_\text{rank}$ and $D(\mu_s,\mu_t)$ need to be minimized according L100-102, but the negative sign in between does the opposite.

**Limitations:**

The work is self-sufficient, and presents good evaluation to convey the effectiveness of the proposed method. Although, I feel that the scope of this or a follow-up work could be broadened a bit, to also incorporate problems beyond passage ranking into the evaluation.

---

> ### Author Rebuttal · Authors · 2023-08-09
>
> We thank the reviewer for the time, comments, and support! We hope that the following addresses the concerns:
>
> - **“results on diverse ranking tasks… like fine-grained computer vision”**
>
> 	Thank you for the suggestion, and it would be indeed interesting to see whether ListDA would be useful in (vision) tasks that reframe fine-grained classification as ranking. Regarding task diversity, in Appendix D, we also included experiments on the Yahoo! Learning to Rank (LETOR) Challenge v2.0 dataset, a web search ranking task on numerical data. The results are consistent with those from the passage reranking experiments and also support our findings in the main sections.
>
> - **“Is there a typo in L98?”**
>
> 	Thank you for pointing this out! It will be fixed in the revision.

---

> > ### Comment · Reviewer_3wjd · 2023-08-19
> > **Response**
> >
> > I thank the authors for addressing my questions, and will keep my rating unchanged.

---

### Author Rebuttal · Authors · 2023-08-09

We thank all reviewers for their time and the remarks!

- **Figure on item-level alignment vs. list-level alignment.**

	We created a new figure (in the attached PDF) to better illustrate their difference as well as the point that the latter is a stronger requirement than the former. This figure will be included in our revision.

- **(Reviewers 3wjd and nxwQ) “the use of transformer discriminator instead of MLP [for ListDA]” and “unconvinced that this improvement is coming from the list-wise invariance and not from the improvement in discriminator architecture”**

	First, we discuss the justifications behind this choice, then discuss the results on a new set of experiments where we used a transformer discriminator for ItemDA (ItemDA-transformer), as well as a MLP discriminator for ListDA (ListDA-MLP).

	We used transformers for ListDA not mainly for its representation power, but for its efficiency in modeling sequential and list/set-like data. It is indeed possible to use discriminators with a MLP architecture for ListDA, e.g., one design (ListDA-MLP) would be to concatenate all feature vectors (each $\in\mathbb R^k$) of a list (of length $\ell$) and form a long vector $\in\mathbb R^{\ell\times k}$, and feed it into the MLP. However, its optimization process will be inefficient because the MLP in this design does not recognize the permutation invariance property of rank lists (i.e., swapping two items in a list with the scores does not alter the list), so it will be much slower to train (exponentially in $\ell$, for iterating through all orderings of the concatenation). Therefore, we used a transformer (without positional encoding) because it is invariant to permutations in the input sequence, and is efficient to optimize, since MLP is not appropriate for ListDA.

	Now, the transformer is indeed a more capable model than MLP, but its additional modeling power is only useful on sequential data. Note that each input (unbatched) to the list-level discriminator (for ItemDA) is a list of $\ell$ feature vectors in $\mathbb R^k$ as a sequence, whereas each input to the item-level discriminator (for ListDA) is a single $\mathbb R^k$ vector. Therefore, if we were to use transformer for item-level alignment (ItemDA), since each input is a sequence of length-1, the attention mechanism becomes pointless, and the transformer collapses to a MLP (with skip connections and layernorm)—in this regard, the transformer is as expressive as the MLP for ItemDA (we also tried wider and deeper MLPs in our experiments, but they did not provide additional benefits). The transformer discriminator is only meaningful for ListDA, where the inputs are sequential. This is not to rule out the possibility of better discriminator architectures, but is outside the scope of our work.

- **(Continuing from above) Results on ItemDA-transformer and ListDA-MLP.**

	We ran additional experiments (ablations) to show that **using a transformer discriminator for ListDA provides no additional benefits**, where we see that ItemDA-transformer has the same performance as ItemDA-MLP on all datasets:

	| Robust04 | MAP | MRR@10 | NDCG@5 | NDCG@10 | NDCG@20 |
	| --- | --- | --- | --- | --- | --- |
	| ItemDA-MLP | 0.2822 | 0.8037 | 0.5822 | 0.5396 | 0.4922 |
	| ItemDA-transformer | 0.2848 | 0.8018 | 0.5851 | 0.5406 | 0.4982 |
	| ListDA | 0.2901 | 0.8234 | 0.5979 | 0.5573 | 0.5126 |

	| TREC-COVID | MAP | MRR@10 | NDCG@5 | NDCG@10 | NDCG@20 |
	| --- | --- | --- | --- | --- | --- |
	| ItemDA-MLP | 0.3086 | 0.9080 | 0.8276 | 0.8142 | 0.7697 |
	| ItemDA-transformer | 0.3088 | 0.8907 | 0.8287 | 0.8069 | 0.7691 |
	| ListDA | 0.3184 | 0.9335 | 0.8693 | 0.8412 | 0.7985 |

	| BioASQ | MAP | MRR@10 | NDCG@5 | NDCG@10 | NDCG@20 |
	| --- | --- | --- | --- | --- | --- |
	| ItemDA-MLP | 0.4781 | 0.6383 | 0.5315 | 0.5343 | 0.5604 |
	| ItemDA-transformer | 0.4739 | 0.6420 | 0.5256 | 0.5312 | 0.5567 |
	| ListDA | 0.5191 | 0.6666 | 0.5639 | 0.5714 | 0.5985 |

	To show that **MLP discriminator is inappropriate for ListDA**, we used the concatenation strategy described above and applied it on the Yahoo! LETOR dataset. We used two training strategies: in ListDA-MLP-1, both the ranking model and the discriminator are updated in every step (with gradient reversal), and in ListDA-MLP-10, the model is updated every 10 discriminator update steps.

	The results are as follows. Because of the difficulty in optimizing ListDA-MLP mentioned above (the complexity is exponential in list-size, and the list size in Yahoo! LETOR dataset can be as large as 139), it is hard to maintain the discriminator to near-optimality (which is required in adversarial training), and this suboptimality leads to negative transfer. Note that ListDA-MLP-10 > ListDA-MLP-1 because the former discriminator updates more frequently (also takes ten times as long to train), suggesting that the negative transfer is due to inefficiencies in optimization.

	| Yahoo! LETOR | MAP | MRR@10 | NDCG@5 | NDCG@10 | NDCG@20 |
	| --- | --- | --- | --- | --- | --- |
	| ItemDA | 0.5315 | 0.6717 | 0.7402 | 0.7708 | 0.8255 |
	| ListDA-transformer | 0.5370 | 0.6771 | 0.7442 | 0.7735 | 0.8269 |
	| ListDA-MLP-1 | 0.4265 | 0.5190 | 0.6385 | 0.6958 | 0.7740 |
	| ListDA-MLP-10 | 0.5041 | 0.6480 | 0.7157 | 0.7508 | 0.8117 |

---

### Decision · Program_Chairs · 2023-09-21

**Decision:**

Accept (spotlight)

**Comment:**

The paper presents a novel method to learn domain-invariant representations for domain adaptation on ranking problems, which is well received by all reviewers (2x accept, 1x weak accept, 1x borderline accept). In particular, it is well-written and motivated, and addresses an interesting and practically important problem. Moreover, the work establishes a new generalization bound for domain adaptation for ranking, and achieves good empirical results.

The rebuttal has addressed most of the initial concerns raised by the reviewers by providing additional justification for the transformer-based discriminator and clarification. Therefore, the AC recommends acceptance of the paper and encourages the authors to revise the paper according to the rebuttal discussion.